# Differentiating interfacial water structures via alkali metal cation promotor for $H_2O_2$ electrosynthesis in acid

Yifei Wang [1] ✉, Peiyang Duan[1], Yingqi Liao[2], Hao Wang[1], Beibei Li[1], Hangyuan Zhang[3], Hao Yang [2] ✉, Tao Cheng [2] ✉ & Jingyu Sun [3] ✉

Electrocatalytic oxygen reduction reaction (ORR) for $H_2O_2$ production represents a sustainable alternative route to the energy-intensive anthraquinone process. Nevertheless, under industrially-relevant acidic conditions, excessive protons at the reaction interface exacerbate low $H_2O_2$ selectivity and severe $H_2O_2$ reduction. Herein, we propose a universal alkali metal cation (AMC: $Li^+$, $Na^+$, $K^+$, or $Cs^+$) dosing strategy to markedly boost the acidic $H_2O_2$ electrosynthesis. Upon $Cs^+$ addition, $2e^-$ ORR selectivity increases from 20% to 80%, concurrently suppressing an $H_2O_2$ reduction current by 50% and achieving an $H_2O_2$ production rate of 9.2 mol $g^{-1}$ $h^{-1}$ at 500 mA $cm^{-2}$. Microelectrode hydrogen evolution measurements witness impeded proton diffusion in AMC-dosed acidic electrolytes, directly restricting proton supply to catalytic active sites. In situ spectroscopic analysis combined with molecular dynamics simulation demonstrate AMCs help reconfigure interfacial water networks via cation hydration shells, thereby disrupting proton-hopping pathways. The efficacy trend ($Li^+$ <$Na^+$ <$K^+$ <$Cs^+$) originates from distinct cation-specific interfacial water restructure, delivering mechanistic insights into cation-promoted selective $H_2O_2$ electrosynthesis in acidic media.

Hydrogen peroxide ($H_2O_2$) has emerged as a green and versatile oxidant with vast application potentials in wastewater treatment, organic synthesis and chemical technology[1–7]. Its on-site electrosynthesis via the two-electron oxygen reduction reaction ($2e^-$ ORR) with enriched choices of electrocatalysts[8], offers an appealing alternative to the energy- and waste-intensive anthraquinone process[9,10]. Nevertheless, the practical implementation at industrially-relevant current densities is hindered by activity and stability limitations, primarily due to competing $4e^-$ reduction pathways and $H_2O_2$ decomposition[11–14].

Beyond deliberate electrocatalysis, the fundamental chemistry of aqueous interfaces has garnered intense interest for its role in spontaneous chemical transformations[15–18]. Notably, the purported spontaneous formation of $H_2O_2$ has sparked significant debate. While early reports emphasized remarkable accelerations at the air-water interface[19–23], a growing body of evidence, including rigorous work from the Mishra group, suggests that such phenomena might be more pronounced at solid–water interfaces[24–27]. This ongoing controversy underscores that the intrinsic drivers of reactivity in confined aqueous environments remain a fundamental question in interfacial science. Resolving this question is critical not only for understanding natural processes but also for the rational manipulation of electrolyte–catalyst interfaces.

[1]National Engineering Laboratory for Advanced Municipal Wastewater Treatment and Reuse Technology, Key Laboratory of Beijing for Water Quality Science and Water Environment Recovery Engineering, Beijing University of Technology, Beijing, China. [2]Institute of Functional Nano & Soft Materials, Jiangsu Provincial Key Laboratory for Carbon-Based Functional Materials & Devices, Joint International Research Laboratory of Carbon-Based Functional Materials and Devices, Soochow University, Suzhou, China. [3]College of Energy, Soochow Institute for Energy and Materials Innovations, Key Laboratory of Advanced Carbon Materials and Wearable Energy Technologies of Jiangsu Province, Soochow University, Suzhou, China. ✉e-mail: wangyifei@bjut.edu.cn; haoyang@suda.edu.cn; tcheng@suda.edu.cn; sunjy86@suda.edu.cn

Electrolyte engineering, particularly the introduction of alkali metal cations (AMCs: Li[+], Na[+], K[+], or Cs[+])[28–30], has emerged as a promising strategy to steer interfacial chemistry toward $H_2O_2$ electrosynthesis in acid[31,32]. AMCs are generally understood to mitigate the local proton activity, thereby stabilizing the key *OOH intermediate and inhibiting $H_2O_2$ decomposition. Particularly using metal-free carbon-based electrocatalysts, Zhang et al. observed that $H_2O_2$ production rates can be improved by an AMC-induced shielding effect, yet were relatively unaffected by the AMC types[33]. A recent study indicated that a graphitic carbon electrode exhibited an increase in 2e[−] ORR selectivity following the trend of Cs[+] > K[+] > Na[+] > Li[+], where different cations would interfere in the *OOH binding and influence the local electric field[34]. It is interesting to note that the AMC roles in acidic 2e[−] ORR manifest inconsistent trends. In further contexts, the cation effect is effective across other catalyst types apart from carbonaceous materials. For instance, Cao et al. found the $H_2O_2$ production followed the trend of K[+] > Na[+] > Li[+] ~ Cs[+] with the employment of earth-abundant TiC catalyst, which was dictated by the hydrated ionic radius of AMCs[35]. These disparities imply that the dominant mechanism might be sensitive to specific conditions, such as catalyst identity, cation concentration and operating current density. A coherent picture of how AMCs reshape the interfacial environment in acidic 2e[−] ORR is still lacking, especially in the realm of prevailing transition metal-involved carbon electrocatalysts (e.g., metal-carbon nanotube, M-CNT).

Inspired by these considerations, we pay attention to a previously underappreciated role of AMCs that lies in their specific capability of reconfiguring the interfacial water to varied degrees. Such an interfacial water restructure governs the kinetics of proton transport to the active site. By employing Co-CNT as a model catalyst and AMC concentration series (0.1–0.5 M), we observe a clear cation-dependent selectivity trend (Li[+] < Na[+] < K[+] < Cs[+]) that correlates directly with spectroscopic signatures of interfacial water reconfiguration, as captured by in situ attenuated total reflectance surface-enhanced infrared absorption spectroscopy (ATR-SEIRAS). Ab-initio molecular dynamics (AIMD) simulations provide strong evidence that AMCs promote the formation of less hydrogen-bonding (H-bonding) or free water molecules. This would markedly elevate the proton transfer energy barrier to impede $H_2O_2$ reduction, which is confirmed by microelectrode measurements. The varying influences of different AMCs stem from their distinct capabilities to perturb the H-bond network, associated with their hydration properties and coordination strengths. Such a perspective is essentially different from the general recognition of AMC roles in this field (Fig. 1). Leveraging this insight, we achieve a considerable improvement in acidic $H_2O_2$ selectivity and a production rate of 9.2 mol g$^{-1}$ h$^{-1}$ at 500 mA cm$^{-2}$ with Cs[+] additives, readily enabling efficient pollutant degradation and bacterial disinfection under practical application scenarios.

## Results

### Investigating AMC-dependent proton transport suppression

It is a general recognition that proton concentration at the electrode surface critically governs acidic 2e[−] ORR process[33,36,37]. We employed gold microelectrode measurement to probe the influence of AMCs upon the proton delivery toward the electrode. This could be achieved by evaluating the diffusion-limited current density of the hydrogen evolution reaction (HER)[38,39]. First, linear sweep voltammograms (LSVs) were recorded on a 5-mm diameter Au rotating disk electrode in

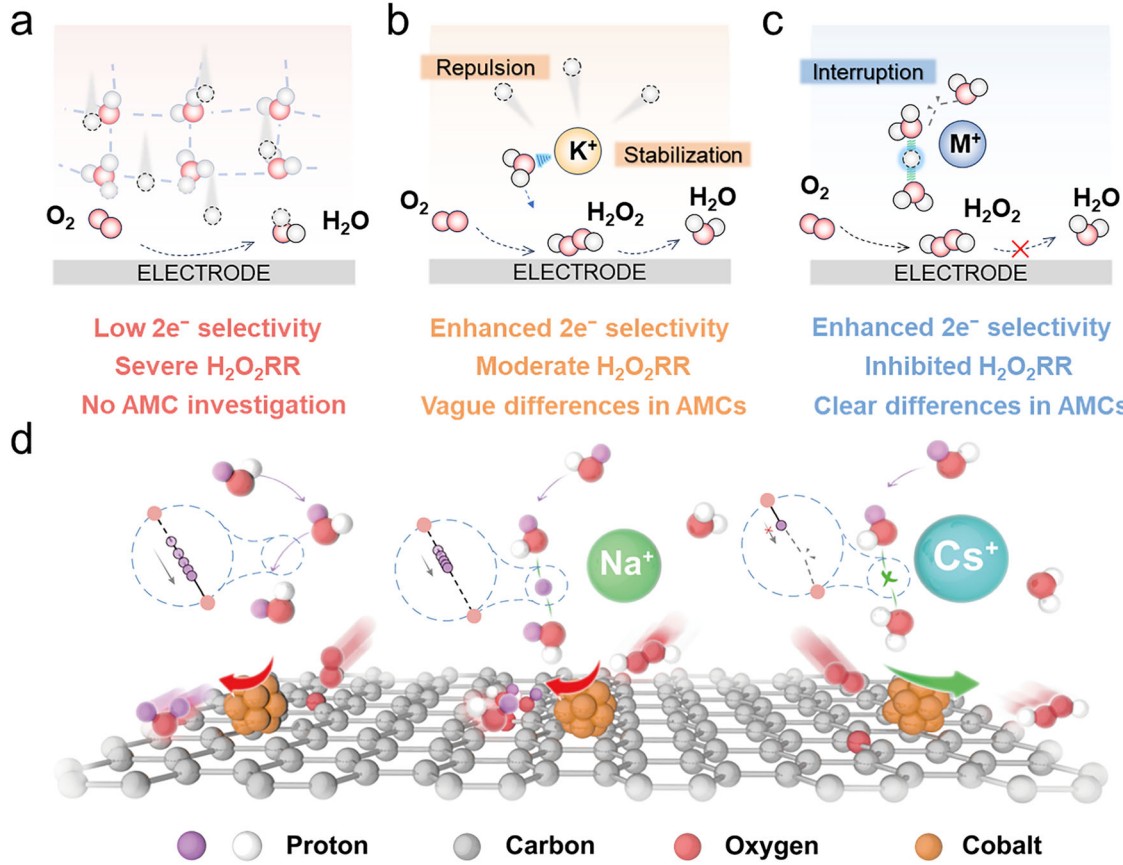

**Fig. 1 | Schematic illustrations of AMC regulating the production of $H_2O_2$.** The ORR pathways for $H_2O_2$ production in **a** conventional acidic electrolyte, **b** typical AMC-modified acidic electrolyte, and **c** our present AMC-dosed acidic electrolyte. **d** Cation-dependent promotion mechanisms of the 2e[−] ORR in AMC-dosed acidic electrolyte presented in this work.

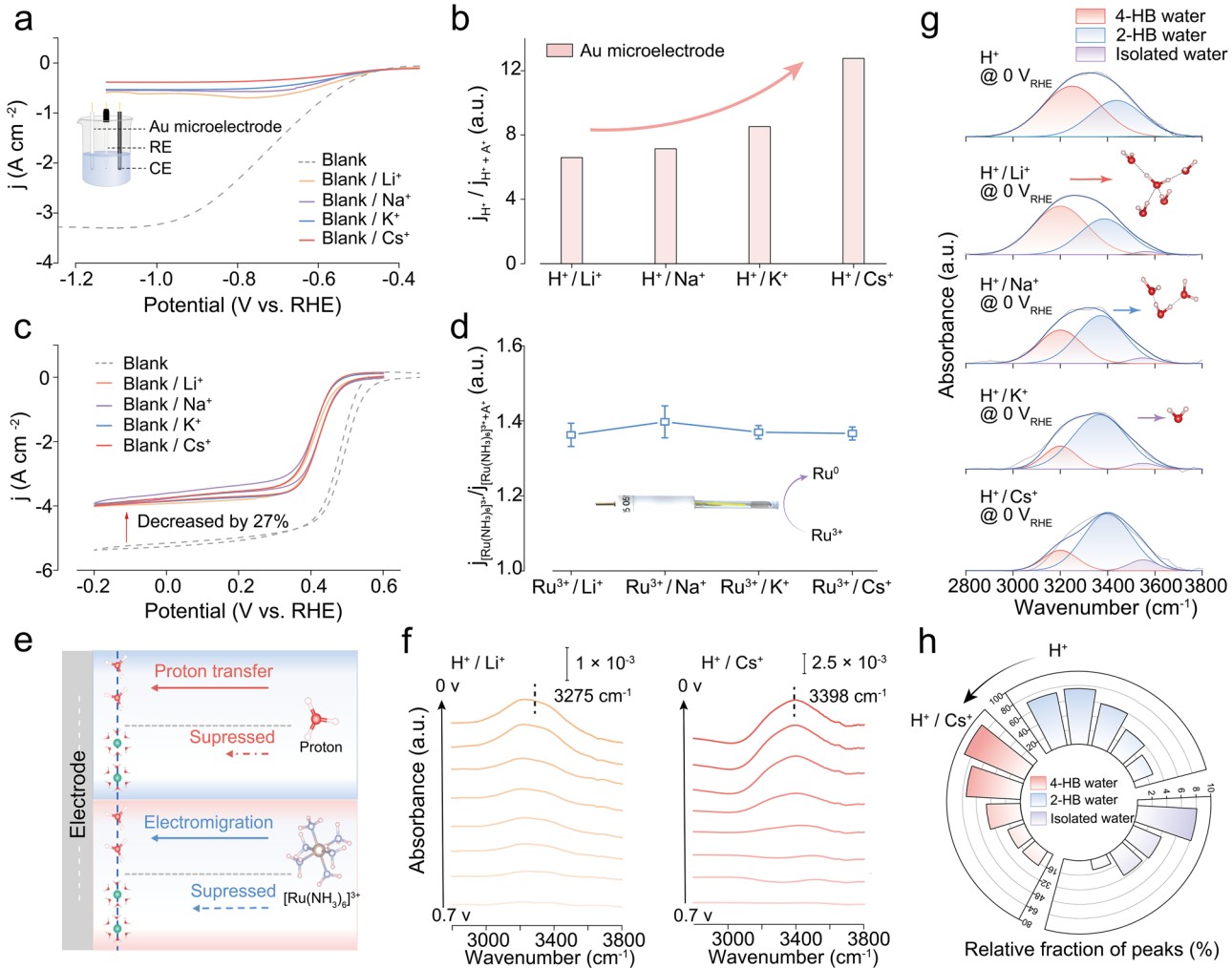

**Fig. 2 | Investigation on the impact of AMC on the structure of the interface water. a** HER polarization curves for Au microelectrodes in Ar-saturated 0.5 M H₂SO₄ (dashed) and 0.5 M H₂SO₄ + 0.15 M A₂SO₄ (solid) at 0.02 V s⁻¹. The inset shows the experimental setup from a to d. **b** Diffusion-limited current density ratios for 1 M H⁺ and 1 M H⁺ + 0.3 M A⁺ (A = Li, Na, K, Cs). **c** Steady-state CVs of Au microelectrodes in Ar-saturated 10 mM [Ru(NH₃)₆]Cl₃ without/with 0.15 M A₂SO₄ at 0.02 V s⁻¹. **d** Diffusion-limited current ratios for 10 mM [Ru(NH₃)₆]³⁺ without/with 0.3 M A⁺. The inset shows the reduction reaction of Ru on the surface of an Au microelectrode. e Mass transfer schematics for H⁺ and [Ru(NH₃)₆]³⁺ with/without A⁺

(A = Li, Na, K, Cs). **f** The potential-dependent OH stretching features of in situ ATR-SEIRAS spectra in electrolytes (0.5 M H₂SO₄ and 0.5 M H₂SO₄ + 0.15 M Cs₂SO₄). **g** Deconvoluted OH-stretching region showing three components: 3550 cm⁻¹ (isolated water), 3400 cm⁻¹ (2-HB water), and 3200 cm⁻¹ (4-HB water). The insets show molecular structure models of different H-bonded water configurations. The balls in (**e**, **g**) represent H (white), O (red), N (blue), and Ru (gold). **h** Cation-dependent fractional contributions of 3550/3400/3200 cm⁻¹ bands. The potential is not iR corrected. Data in d is presented as mean ± s.d. (n ≥ 3). Source data for Fig. 2 are provided as a Source Data file.

a test solution of 0.5 M H₂SO₄ containing 0.3 M Cs⁺ at rotation rates of 400, 900, 1600, and 2500 rpm. We observed that the HER current was entirely mass-transport-limited by proton diffusion (Supplementary Fig. 1), confirming the validity of our measurement methodology. In a bare acidic solution, a feature diffusion-limited current plateau of ~3.3 A cm⁻² was attained on the Au microelectrode. Upon the AMC dosage, such a current markedly decreased to below 0.6 A cm⁻² (Figs. 2a, b; Supplementary Figs. 2, 3). Moreover, the extent of current suppression exhibited a strong dependence on the AMC types. Li⁺ exerted the weakest suppression effect, whereas Cs⁺ produced the most pronounced suppression, reaching a maximum factor of 13-fold. It is further noted that there existed a concentration-dependent variation of this suppression effect (Supplementary Figs. 4, 5). The optimized AMC concentration refers to a key interfacial reaction engineering of the balance between suppressing parasitic pathways and maintaining sufficient reaction kinetics. Considering the trade-off between proton availability and reaction kinetics, a medium

concentration of 0.3 M was selected to assess the AMC concentration effect on the H₂O₂ yield (Supplementary Fig. 6).

Gu *et al*. proposed that AMCs could hamper the electromigration of protons in solution via an electric field shielding fashion[38,40]. To examine the electric field modulation on proton transport in our system, we employed [Ru(NH₃)₆]³⁺ ion as a model redox probe and measured the dependence of its diffusion rate on AMCs[39]. Our findings indicated that AMC dosage enabled the suppression of [Ru(NH₃)₆]³⁺ electromigration, but marginal changes in the suppression effect were detected with respect to the dosed AMC types, all yielding an inhibition factor of ~1.38 (Fig. 2c, d). Additionally, this effect did not exhibit further enhancement with increasing Cs⁺ concentration (Supplementary Fig. 7), which cannot account for the ~13-fold suppression factor (Fig. 2e). To exclude potential interference from other variables, the pH value, ionic conductivity and dissolved oxygen in different electrolytes were probed. As shown in Supplementary Figs. 8, 9, the AMC dosage did not significantly alter the values, indicating such

parameters were unlikely to be the primary drivers behind the observed cation-dependent trend. These results motivated us to investigate the restructuring of interfacial H-bonding water networks in the presence of AMCs.

To elucidate the influence of AMCs on the interfacial water structures, we carried out in situ ATR-SEIRAS measurements (Supplementary Fig. 10). The SEIRAS signals were acquired at the catalyst-electrolyte interface in 0.5 M $H_2SO_4$ containing $Li^+$, $Na^+$, $K^+$, or $Cs^+$. The spectra clearly exhibit the O−H stretching vibration band of interfacial water (3000−3600 cm$^{-1}$) (Fig. 2f; Supplementary Fig. 11)[41]. The intensity displays apparent potential dependence, indicating that the signals primarily originated from the first few layers of water molecules adjacent to the electrode surface. It is noted that the O−H stretching vibration band under different potentials in bare acid and $Li^+$-containing system was predominantly located at ~3270 cm$^{-1}$, affording a redshift of ~130 cm$^{-1}$ as compared to that in the $Cs^+$-containing system (~3400 cm$^{-1}$). Such a shift implied that interfacial water molecules formed strong H-bond interactions at the electrode surface in the presence of small AMCs, which is analogous to the scenario in bulk water. In contrast, electrolytes containing large AMCs induced a reduction in the number of H-bonds formed by interfacial water molecules, resulting in a higher O−H stretching vibration frequency.

In detail, the O−H stretching band was deconvoluted into three components centered at 3550, 3400, and 3200 cm$^{-1}$ (Fig. 2g)[42–48]. These signals could be assigned to isolated water molecules, 2-coordinated H-bonded water (2-HB), and 4-coordinated H-bonded water (4-HB), respectively, with their cation-dependent relative contributions plotted in Fig. 2h. As such, large AMC significantly enhanced the population of 2-HB and isolated water species, suggestive of pronounced disruption of the H-bond network within the interfacial water layer. The extent of this disruption also afforded the order of $Li^+ < Na^+ < K^+ < Cs^+$, which is consistent with the trend observed in proton transport suppression. Collectively, these findings underline the importance of AMC dosage in disrupting the H-bond network and altering the interfacial H-bonded water structures.

## Theoretical insight into AMC-suppressed proton transport

Computational simulations were carried out to validate the experimental observations. Prior to this, the prepared catalyst underwent a full spectrum of characterizations, including electron microscopy (Supplementary Figs. 12–14), X-ray photoelectron spectroscopy (Supplementary Figs. 15–17), Fourier transform infrared spectroscopy (Supplementary Fig. 18), X-ray diffraction (Supplementary Fig. 19), and synchrotron radiation-based analysis (Supplementary Figs. 20–23; Supplementary Table 1). The introduction of potassium thiocyanate (KSCN) into the electrolyte is known to selectively bind and poison transition metal sites. As demonstrated in Supplementary Fig. 24, the addition of 10 mM KSCN leads to a pronounced decrease in disk current density, suggesting cobalt acts as the active site for the 2e$^-$ ORR. These unambiguously revealed the morphological, chemical and electronic information of catalysts.

We constructed an orthorhombic supercell model incorporating 1080 explicit water molecules. Protons (H$^+$) and AMCs (Li$^+$, Na$^+$, K$^+$, Cs$^+$) were then introduced into such an aqueous environment to replicate the experimental electrolyte conditions (Supplementary Fig. 25). The system underwent 200 ps of pre-equilibration by NVT simulations to generate physically realistic initial configurations, followed by extensive equilibration by 2 ns NPT and 2 ns NVT. After equilibration, quantitative analysis of H-bond populations across the different systems revealed the sequence of $Li^+ < Na^+ < K^+ < Cs^+$, where the Cs$^+$-containing system exhibited the lowest H-bond density per unit volume (Supplementary Fig. 26; Supplementary Table 2). Subsequently, path integral molecular dynamics (PIMD) simulations were performed to probe the proton transport in the presence of AMCs (Fig. 3a). Quantification of proton conduction pathways reveals that AMC suppresses long pathways while promoting short trajectories, with the trend remaining $Li^+ < Na^+ < K^+ < Cs^+$ (Fig. 3b, c; Supplementary Figs. 27–29). Moreover, larger AMCs coordinating more water molecules markedly reduce the number of H-bonds, as reflected in the radial distribution functions (RDFs) (Supplementary Fig. 30). These results collectively indicate that large-sized AMCs impede proton transport.

Gomez et al. elucidated a three-step mechanism for aqueous proton transport, governed by sequential H-bond exchange events: (i) the reduction in proton-acceptor coordination to facilitate proton transfer; (ii) the rate-limiting step involving enhanced proton-donor coordination to prevent back-transfer; and (iii) the dynamic equilibrium between Eigen and Zundel configurations[49–51]. Based upon the theoretical framework and in situ ATR-SEIRAS analysis, three distinct proton-donating water species were introduced as reference configurations, encompassing 4-HB water, 2-HB water, and isolated (free) water, with 4-HB water serving as the universal proton acceptor[52]. Consequently, donor-acceptor pairs were categorized into three types: Type I: 4-HB water (donor) → 4-HB water (acceptor); Type II: 2-HB water (donor) → 4-HB water (acceptor); Type III: Free water (donor) → 4-HB water (acceptor).

Density functional theory (DFT) calculations were performed to elucidate how different H-bond configurations affect proton transfer. In the Type I system, the proton transfer from donor to acceptor proceeded in a stable manner. At the IM1 intermediate state, the proton-donor water distance ratio reached 1%, indicating the proton remains associated with the donor. At the IM6 state, this ratio reversed with the proton-acceptor water distance attaining 9%, demonstrative of successful proton transfer to the acceptor (Fig. 3d; Supplementary Fig. 31). In contrast, Type II systems exhibited comparatively slower proton transfer kinetics. The proton-donor water distance ratio merely shifted from 4% at IM1 to 82% at IM6 (Fig. 3d; Supplementary Fig. 32). During the transfer from IM1 to IM3, the proton reached the midpoint between the donor and acceptor (50%) more rapidly than in the Type I system. However, its mobility decreases significantly along the pathway from IM4 to IM6. After advancing to 82% of the donor–acceptor distance at IM5, the proton remained at this position in IM6. This behavior indicates that in the Type II system, the proton exhibited a stronger tendency to form and maintain a Zundel-like configuration ($[H_2O \cdot H \cdot OH_2]^+$)[51].

Paradoxically, while the Zundel state is a key intermediate, efficient proton transfer relies on the dynamic transition between this state and the intrinsic Eigen-type configuration. Stabilization of the Zundel state in the Type II system introduces a kinetic bottleneck, which likely suppresses overall proton transfer. Furthermore, the complete proton transfer in the Type II system shows an energy barrier of 0.011 eV, about twice that in the Type I system (0.005 eV), indicating hindered proton mobility considering the intrinsic uncertainty of DFT calculations (Fig. 3e). The kinetic impediment likely originated from reduced H-bond coordination of the donor water molecule in Type II systems, which stabilized the symmetric Zundel intermediate during proton transfer. Additionally, H-bond network deformation was observed during the 1-HB to 4-HB transition, manifested through the formation of 2-HB donors and 3-HB acceptors, suggesting that direct 1-HB → 4-HB transfer faced elevated energy barriers (Supplementary Fig. 33). Consequently, the experimentally observed suppression of proton transfer by AMCs can be attributed to the cation-dependent restructuring of interfacial water molecules.

## AMC-modulated *OOH binding

The kinetics of *OOH intermediate formation, desorption, and dissociation are the key to governing the 2e$^-$ ORR selectivity toward high-rate $H_2O_2$ electrosynthesis[53–55]. To elucidate the origin of high 2e$^-$ pathway selectivity over Co-CNT electrocatalysts under varying AMC dosages, different types of reaction models with respect to pure H$^+$, H$^+$ + Li$^+$, H$^+$ + Na$^+$, H$^+$ + K$^+$, and H$^+$ + Cs$^+$ media were constructed. Upon

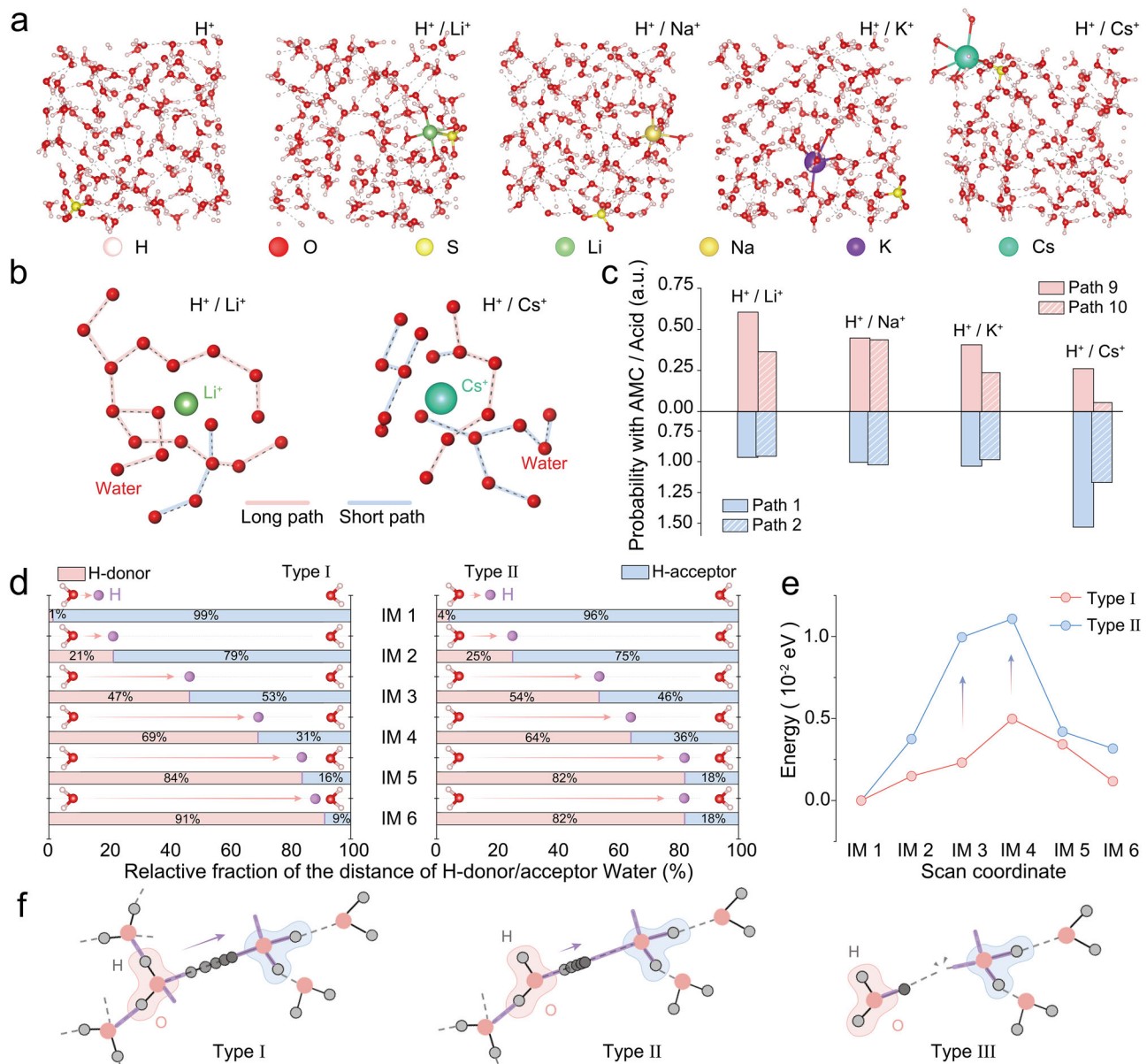

**Fig. 3 | Theoretical simulation of the influence of AMC on interfacial proton transfer. a** Molecular dynamics model employed to quantify H-bond network connectivity utilized cubic simulation boxes with a side length of 7 Å. The system was populated with hydronium ions, sulfate anions, water molecules, and AMCs (Li$^+$/Na$^+$/K$^+$/Cs$^+$) to simulate the desired electrochemical interface environment. **b** Hydrogen bonding network connectivity extracted via graph theory for the acid solutions with Li$^+$ (left panel) and Cs$^+$ (right panel). **c** Relative probabilities of path number of hydrogen bonds in acid solution with AMCs to pure acid solution. Water networks in pure acid solution exhibit higher connectivity than those with AMCs in the PIMD simulation. **d** The fractional distances from the proton to the donor water and to the acceptor water during proton transfer. **e** Free energy profile of the transition state for the proton transfer pathway from a 4-HB/2-HB configuration to a 4-HB configuration. **f** Schematic illustrating the key influence of different AMCs. Source data for Fig. 3 are provided as a Source Data file and Supplementary Data 1.

evaluating various adsorption sites for key ORR intermediates (*OOH, *O, and *OH), the most stable configurations were identified (Fig. 4a; Supplementary Figs. 34–39). The reaction initiated with proton-coupled electron transfer (PCET) to form *OOH adsorbed on Co sites. The negative free energy change implied the thermodynamic favorability of *OOH formation. Subsequently, *OOH could either undergo further hydrogenation via the 4e⁻ pathway to form *O and H$_2$O, or desorb as H$_2$O$_2$ via the 2e⁻ pathway.

As depicted in Fig. 4b, at 0.7 V vs. RHE (the equilibrium potential of H$_2$O$_2$/H$_2$O), the energy barrier for the conversion of O$_2$ to H$_2$O$_2$ in the Cs$^+$-dosed system remained the lowest (−1.05 eV), reaching nearly half of that observed in the bare acid system (−2.07 eV). A comparison

of the energy barriers across different AMC systems revealed a progressive decrease (Li$^+$: −1.33 eV, Na$^+$: −1.22 eV, K$^+$: −1.16 eV). This trend indicates that the H$_2$O$_2$ generation required overcoming a higher energy barrier in the pure acid system, while the incorporation of AMCs mitigated this issue by reducing the transport of protons to the surface. Furthermore, H$_2$O$_2$ production is progressively facilitated in the order Li$^+$ < Na$^+$ < K$^+$ < Cs$^+$, in close agreement with experimental observations.

The tendency of the *OOH intermediate to desorb/dissociate is equally crucial to influence H$_2$O$_2$ production[1,56,57]. In bare acidic systems, excessive protons promoted dissociation of adsorbed *OOH intermediates. The energy barrier for *O formation (1.58 eV) was lower

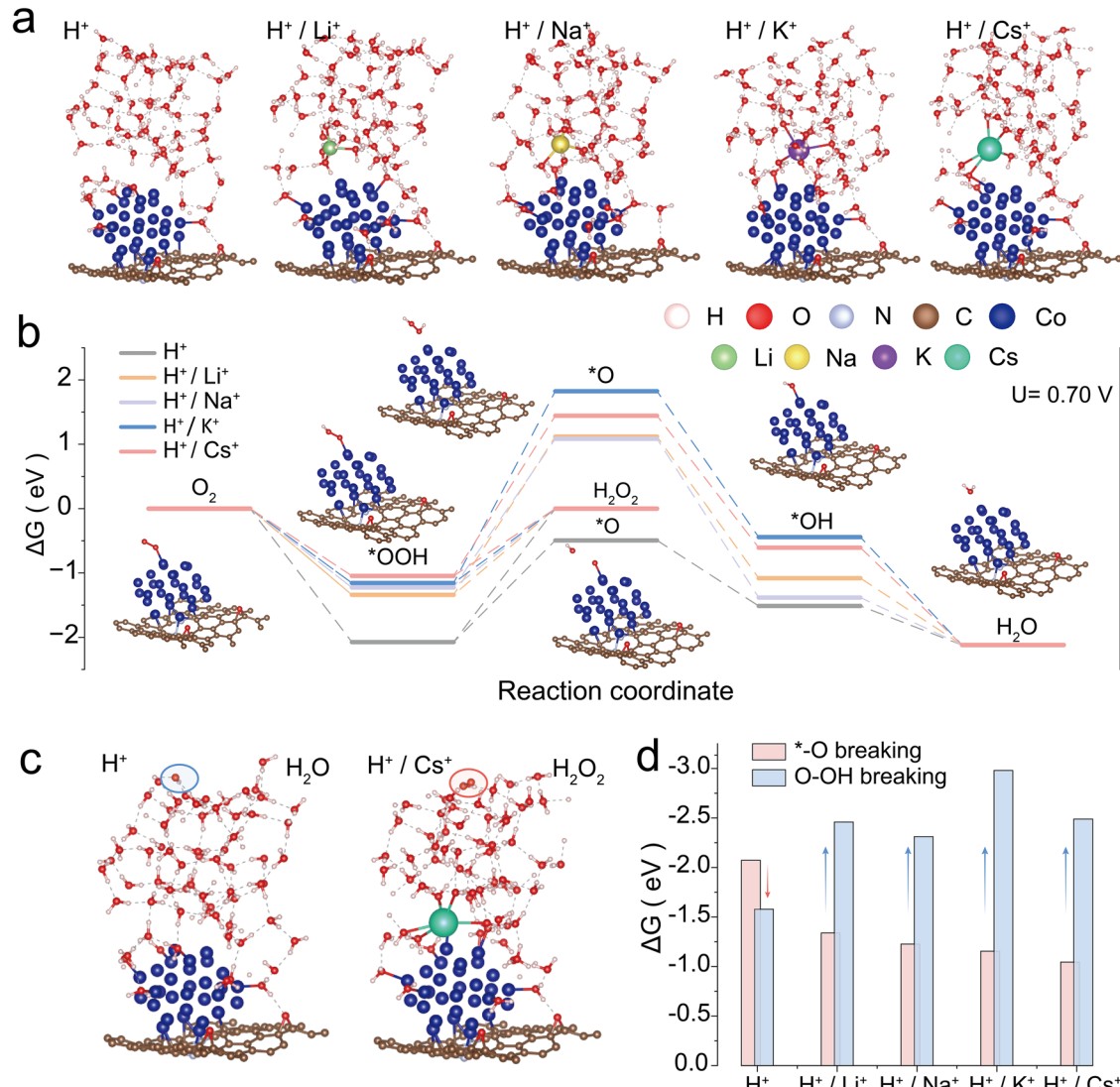

**Fig. 4 | Theoretical simulation of AMC-induced $H_2O_2$ production. a** The optimized computational adsorption configuration model on the Co-CNT catalyst. Calculations were performed to construct free energy diagrams for the hydrogen peroxide electrosynthesis pathway in $A^+$-containing acidic electrolyte (A = Li, Na, K, Cs). **b** Free energy diagrams for the ORR on Co-CNT catalysts were constructed across different electrolyte systems at electrode potentials of 0.7 $V_{RHE}$, encompassing both the 4-electron ($4e^-$) and 2-electron ($2e^-$) reaction pathways. The insets show molecular models of different stages for $H_2O_2$ synthesis. **c** Stable structures of $H_2O$ desorbed intermediate in an acidic electrolyte and $H_2O_2$ desorbed intermediate in a $Cs^+$-containing acidic electrolyte. **d** The plot of overpotential for Co-CNT in various systems. Source data for Fig. 4 are provided as a Source Data file and Supplementary Data 1.

than that for $H_2O_2$ desorption (2.07 eV), favoring the $4e^-$ ORR pathway. In contrast, AMC-dosed systems exhibited substantially higher energy barrier for the *OOH → *O step (1.05 eV with $Cs^+$ compared to 2.49 eV in bare acid), demonstrating that the presence of AMCs suppressed the dissociation of *OOH by limiting the proton availability at the surface. This was a common effect across all AMC-incorporated systems, rendering $2e^-$ pathway and promoting $H_2O_2$ formation. Notably, the $Cs^+$-dosed system showed a pronounced reduction in the $H_2O_2$ formation barrier. The computational trend in AMC efficacy ($Li^+ < Na^+ < K^+ < Cs^+$) was in good agreement with our experimental measurements in ORR selectivity (Fig. 4c, d).

## Electrochemical analysis of AMC impact

The acidic ORR performance of our catalyst was evaluated using a rotating ring-disk electrode (RRDE) system in $O_2$-saturated 0.5 M $H_2SO_4$ containing 0.3 M AMC additive (Fig. 5a). In this sense, $A_2SO_4$ (A: AMC) was deliberately employed. Figure 5b displays LSV curves

collected at a rotation speed of 1600 rpm. The platinum ring electrode was polarized at 1.2 $V_{RHE}$ to oxidize $H_2O_2$ generated on the disk electrode, enabling quantitative analysis of the ORR products. The addition of AMCs significantly delayed the onset potential and lowered the disk current density as compared to the bare electrolyte scenario, which was likely to originate from hindered proton transfer in impeding the HER or $4e^-$ ORR pathways occurring at the catalyst surface[10,58]. The diffusion-limited ring current density, strongly correlated with the $2e^-$ ORR pathway, was obviously enhanced upon AMC addition. The introduction of AMCs managed to restructure the H-bond network and interfacial water configuration, thereby reducing the proton availability at the reaction interface. This effectively suppressed the dissociation of oxygen reduction intermediates, favoring their desorption and augmenting $H_2O_2$ selectivity. The electron transfer numbers ($n$) derived from LSV analysis confirmed that the $Cs^+$-dosed system exhibited the highest $H_2O_2$ selectivity (reaching 80%) and lowest $n$ value (2.5) across a broad potential range of 0−0.6 $V_{RHE}$

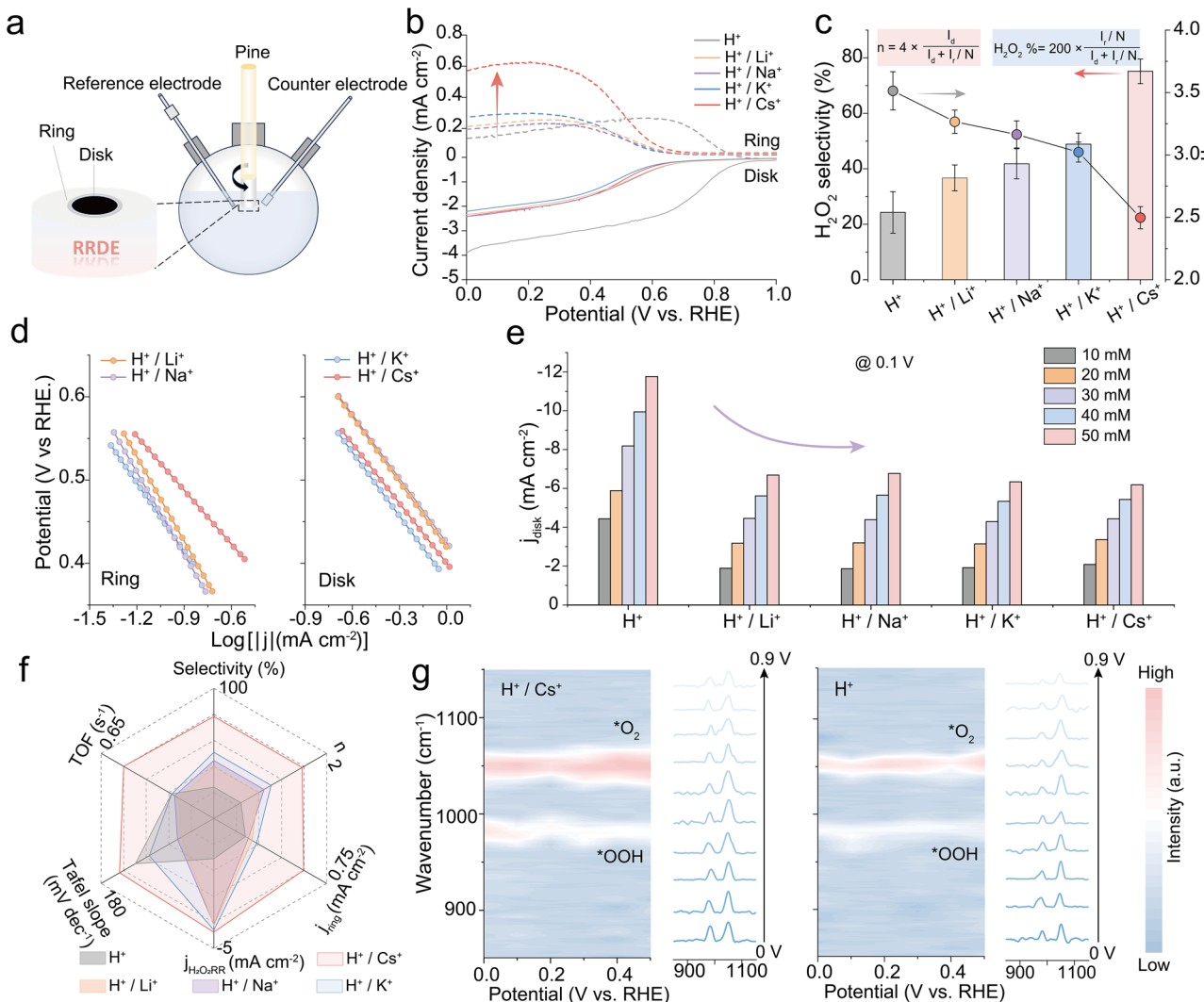

**Fig. 5 | The employed electrochemical RRDE system for the 2e⁻ ORR. a** RRDE configuration schematic for $H_2O_2$ selectivity quantification. **b** LSV and **c** corresponding $H_2O_2$ selectivity/electron transfer numbers ($n$) for Co-CNT in $O_2$-saturated electrolytes: 0.5 M $H_2SO_4$ and 0.5 M $H_2SO_4$ + 0.15 M $A_2SO_4$ (A = Li, Na, K, or Cs). The insets show the calculation formulas for the number of transferred electrons and $H_2O_2$ selectivity. **d** Tafel slopes derived from the LSV data across tested electrolytes. **e** $H_2O_2RR$ current densities for the Co-CNT in $N_2$-saturated 0.5 M $H_2SO_4$ + 0.3 M $A^+$ (A = Li, Na, K, or Cs) containing different concentrations of $H_2O_2$. **f** Multivariate performance radar chart for Co-CNT ORR metrics in different electrolytes. **g** In situ Raman spectra and corresponding contour map derived from ten representative spectra for the Co-CNT in 0.5 M $H_2SO_4$ + 0.15 M $Cs_2SO_4$ (left panel) and 0.5 M $H_2SO_4$ (right panel) during operation. The potential is not iR corrected. Data in (**c**) is presented as mean ± s.d. ($n \geq 3$). Source data for Fig. 5 are provided as a Source Data file.

(Fig. 5c; Supplementary Fig. 40). We observed that the 2e⁻ ORR selectivity in acid followed a monotonically increasing trend from Li⁺ to Cs⁺. Our catalyst showcased a competitive turnover frequency (TOF) value of 0.51 s⁻¹ at 0.5 $V_{RHE}$, which compared favorably with reported electrocatalysts[59,60]. In addition, Tafel slope plots derived from the Koutecky-Levich diffusion equation manifested that the Cs⁺ system exhibited the fastest ORR kinetics (Fig. 5d; Supplementary Fig. 41)[61,62].

In addition, electrochemical double-layer capacitance measurements demonstrate that the catalyst exhibited the highest double-layer capacitance in Cs⁺-dosed electrolyte, suggesting an optimized interfacial environment that promotes reaction activity and electron transfer (Supplementary Fig. 42)[63,64]. RRDE tests in $N_2$-saturated 0.5 M $H_2SO_4$ containing 10-50 mM $H_2O_2$ showed significantly suppressed $H_2O_2$ reduction currents in AMC-dosed electrolytes[65]. The Cs⁺ system yielded the weakest current response for $H_2O_2RR$, particularly at high $H_2O_2$ concentrations (Fig. 5e; Supplementary Fig. 43). This effectively mitigates the thorny issue of excessive $H_2O_2$ reduction caused by high proton concentrations in acidic media. The suppression efficacy of

AMCs on $H_2O_2RR$ followed the trend of Li⁺ < Na⁺ < K⁺ < Cs⁺ (Fig. 5f). Moreover, in situ Raman spectroscopy was employed to monitor interactions between oxygen intermediates and the catalyst in both bare and Cs⁺-modified acidic electrolyte. Within the applied potential range of 0 to 0.9 $V_{RHE}$, characteristic Raman peaks emerged at 983 and 1051 cm⁻¹ in both systems (Fig. 5g), corresponding to *OOH and *$O_2$ species, respectively[14]. Comparative analysis revealed the *OOH signal intensity was markedly enhanced in the Cs⁺-containing electrolyte relative to the bare acid counterpart, suggesting the preferential stabilization of the *OOH intermediate in the presence of Cs⁺. Taken together, these findings corroborate that AMC dosage could impede the proton transfer and accordingly promote the 2e⁻ ORR selectivity for boosted $H_2O_2$ production in acid.

## $H_2O_2$ generation and utilization enabled by AMCs

We first deposited the Co-CNT catalyst at a loading of 0.1 mg cm⁻² onto a gas diffusion electrode (Supplementary Figs. 44, 45). With the AMC promoter in hand, we then employed 0.5 M $H_2SO_4$ + 0.3 M $A_2SO_4$ as the

flowing electrolyte and Pt as the counter electrode. We managed to assemble a three-phase flow cell reactor to evaluate the impact of AMCs in practical $H_2O_2$ production (Supplementary Figs. 46, 47). Under a constant-current operation at 50 mA cm$^{-2}$, Cs$^+$ dosage demonstrated enhanced promotion effects relative to other cations. Both $H_2O_2$ production rate and Faradaic efficiency followed the trend of Li$^+$ <Na$^+$ <K$^+$ <Cs$^+$ (Fig. 6a; Supplementary Figs. 48–50). Notably, the $H_2O_2$ production rate increased substantially when elevating the current densities, particularly at industrially-relevant levels (>300 mA cm$^{-2}$). As shown in Fig. 6b, at 500 mA cm$^{-2}$, the production rate reached 9.2 mol g$^{-1}$ h$^{-1}$. We further assessed the production efficiency and operational stability. Under a constant-current electrolysis at 300 mA cm$^{-2}$ in 0.5 M $H_2SO_4$/0.3 M Cs$^+$ electrolyte (50 mL), $H_2O_2$ rapidly accumulated to 5100 mg L$^{-1}$ within 6 h, outperforming 2000 mg L$^{-1}$ attained in bare $H_2SO_4$ electrolyte (Fig. 6c). Continuously stable $H_2O_2$ generation was maintained for 140 h at 50 mA cm$^{-2}$ (electrolyte flow rate: 6 mL min$^{-1}$), yielding a consistent output concentration of ~5200 mg L$^{-1}$ (Fig. 6d). These results underscore the optimized performance of the Co-CNT catalyst for electrochemical $H_2O_2$ synthesis in AMC-dosed acidic media, competing the state-of-the-art systems (Supplementary Table 3). Note that an analogous effect was also observed under identical reaction conditions when using Fe-CNT or Ni-CNT catalysts (Supplementary Fig. 51).

The on-site generation of $H_2O_2$ in our system is highly attractive for targeted applications such as wastewater treatment. Our foregoing evaluation confirmed that dosed AMC concentrations ranging from 0.1–0.5 M would enhance $H_2O_2$ electrosynthesis, with core investigation at 0.3 M that could be encountered in industrial effluents such as reverse osmosis concentrates. Under such conditions, endogenous ions serve as inherent promoters, eliminating the external additives. Specifically, the electro-synthesized $H_2O_2$ reacts with supplemented Fe$^{2+}$ via an electro-Fenton process, producing hydroxyl radicals (•OH)[66–68]. We selected model contaminant species spanning (i) broad-spectrum dyes (rhodamine B, methyl orange, methylene blue), (ii) aromatic pollutants (phenol, 4-chlorophenol, 4-nitrophenol) and (iii) sulfonamide contaminants (sulfamethoxazole, sulfadiazine, sulfa pyridine) (Supplementary Figs. 52–54). Our protocol achieved high removal efficiencies (>92% within 30 min) across all contaminant types (Fig. 6e). As for methylene blue exhibiting rapid degradation (>95% removal in 5 min), time-resolved UV-vis spectra confirmed progressive decay of featured absorption peaks at 664 nm (Fig. 6f; Supplementary Fig. 55), validating efficient ·OH-mediated mineralization. Furthermore, the electro-synthesized $H_2O_2$ exhibited effective bactericidal properties. Both *E. coli* and *S. aureus* cultures achieved a 100% sterilization rate within 90 min of exposure (Fig. 6g–i).

## Discussion

In summary, this work identifies alkali metal cations (AMCs: Li$^+$, Na$^+$, K$^+$, Cs$^+$) as effective promoters for acidic $H_2O_2$ electrosynthesis over synthesized Co-CNT catalysts. AMC addition markedly enhanced electrocatalytic performance, with efficacy following the order Li$^+$ < Na$^+$ < K$^+$ < Cs$^+$. Notably, Cs$^+$ boosted $H_2O_2$ selectivity from 20% to 80% and achieved a high production rate of 9.2 mol g$^{-1}$ h$^{-1}$ at 500 mA cm$^{-2}$, demonstrating great potential for industrial application. This promoting effect proved universal across Fe-CNT and Ni-CNT catalysts, underscoring its mechanistic generality. Mechanistic studies revealed that interfacial AMCs restructure water networks and disrupt proton-hopping pathways by inhibiting Eigen-Zundel interconversion, thereby reducing proton availability at active sites. Guided by theoretical calculations, we propose that this simultaneously stabilizes the OOH intermediate while increasing the barrier for O formation by 1.37 ± 0.34 eV, thereby suppressing both $H_2O_2$ reduction and four-electron ORR pathway, consistent with experimental observations. The cation-dependent performance trend stems from the distinct capabilities of larger ions like Cs$^+$ to more strongly perturb H-bond

networks. Beyond providing fundamental insights into interfacial proton management, our work might bridge advanced electrocatalysis with sustainable environmental applications by turning salinity from a treatment hurdle into a catalytic advantage.

## Methods

### Materials

Cobalt(II) nitrate hexahydrate (Co(NO$_3$)$_2$·6H$_2$O, 99 wt%), melamine (C$_3$H$_6$N$_6$, 98 wt%), cyanuric acid (C$_3$H$_3$N$_3$O$_3$, 98 wt%), and potassium titanyl oxalate (K$_2$TiO(C$_2$O$_4$)$_2$ or C$_4$H$_2$K$_2$TiO, 99 wt%) were purchased from Shanghai Macklin Biochemical Co., Ltd. Potassium hexacyanoferrate(III) (K$_3$[Fe(CN)$_6$], 99 wt%) was obtained from Aladdin Reagent Co., Ltd. A 5 wt% Nafion 117 solution was supplied by DuPont Co., Ltd. Concentrated sulfuric acid (H$_2$SO$_4$, 98 wt%), nitric acid (HNO$_3$, 68 wt%), and ethanol (C$_2$H$_5$OH) were purchased from Sinopharm Chemical Reagent Co., Ltd. Lithium sulfate (Li$_2$SO$_4$, 99.5 wt%), sodium sulfate (Na$_2$SO$_4$, 99.5 wt%), potassium sulfate (K$_2$SO$_4$, 99.5 wt%), and cesium sulfate (Cs$_2$SO$_4$, 99.5 wt%) were acquired from Xi'an Yizhichen Biotechnology Co., Ltd. All chemicals were used as received without further purification. Carboxyl-functionalized carbon nanotubes (OCNTs) were obtained from Beijing Boyu Gaoke New Material Technology Co., Ltd. These CNTs were synthesized via chemical vapor deposition. Ultrapure water (resistivity: 18.25 MΩ cm$^{-1}$) used throughout the experiments was purified using a Milli-Q system. The Au microdisk electrode was obtained from CH Instruments, Inc.

### Synthesis of Co-CNT

Oxygen-functionalized carbon nanotubes anchored with cobalt clusters (denoted as Co-CNT) were synthesized via a sequential chemical impregnation, pyrolysis, and acid etching process. Initially, OCNTs were immersed in a mixed acid solution of concentrated $H_2SO_4$ (75 mL) and HNO$_3$ (25 mL) and ultrasonicated at room temperature (20 °C) for 30 min to enhance dispersion. The suspension was subsequently refluxed at 80 °C in an oil bath for 1 h. After cooling to room temperature (20 °C), the sample was isolated by centrifugation at 6000 rpm, thoroughly washed with ethanol, and finally dried overnight in a vacuum oven at 60 °C. For cobalt precursor impregnation, cobalt (II) nitrate hexahydrate (Co(NO$_3$)$_2$·6H$_2$O, 2 mM), cyanuric acid (1.02 g), and melamine (1 g) were dissolved in a 100 mL 1:1 (v/v) ethanol/ultrapure water mixture. This solution was ultrasonicated for 30 min followed by continuous stirring for 24 h. The resulting solid was collected by centrifugation at 6000 rpm, washed three times with ethanol, and vacuum-dried. The dried precursor powder was fine ground and then subjected to pyrolysis in a tube furnace under a continuous N$_2$ flow at 900 °C for 3 h, employing a heating rate of 5 °C min$^{-1}$.

### Characterizations

Powder X-ray diffraction (XRD) patterns were acquired on a Bruker D8 Advance diffractometer using Cu Kα radiation (λ = 1.5418 Å) operated at 40 kV and 40 mA. High-angle annular dark-field scanning transmission electron microscopy (HAADF-STEM) images were obtained using a JEOL JEM-2100F field-emission electron microscope operating at an accelerating voltage of 200 kV. Elemental composition analysis was performed by inductively coupled plasma atomic emission spectroscopy (ICP-AES) on a Thermo Fisher Scientific iCAP 6300 spectrometer. X-ray photoelectron spectroscopy (XPS) measurements were conducted on a Thermo Scientific ESCALAB 250Xi spectrometer with monochromatic Al Kα X-ray radiation (1486.6 eV). Fourier transform infrared (FT-IR) spectra of the catalysts were acquired using a Nicolet Nexus 670 spectrometer employing the KBr pellet technique.

### In situ ATR-SEIRAS

Electrochemical ATR-SEIRAS measurements were performed using a Bruker INVENIO-R FTIR spectrometer equipped with a liquid-nitrogen-

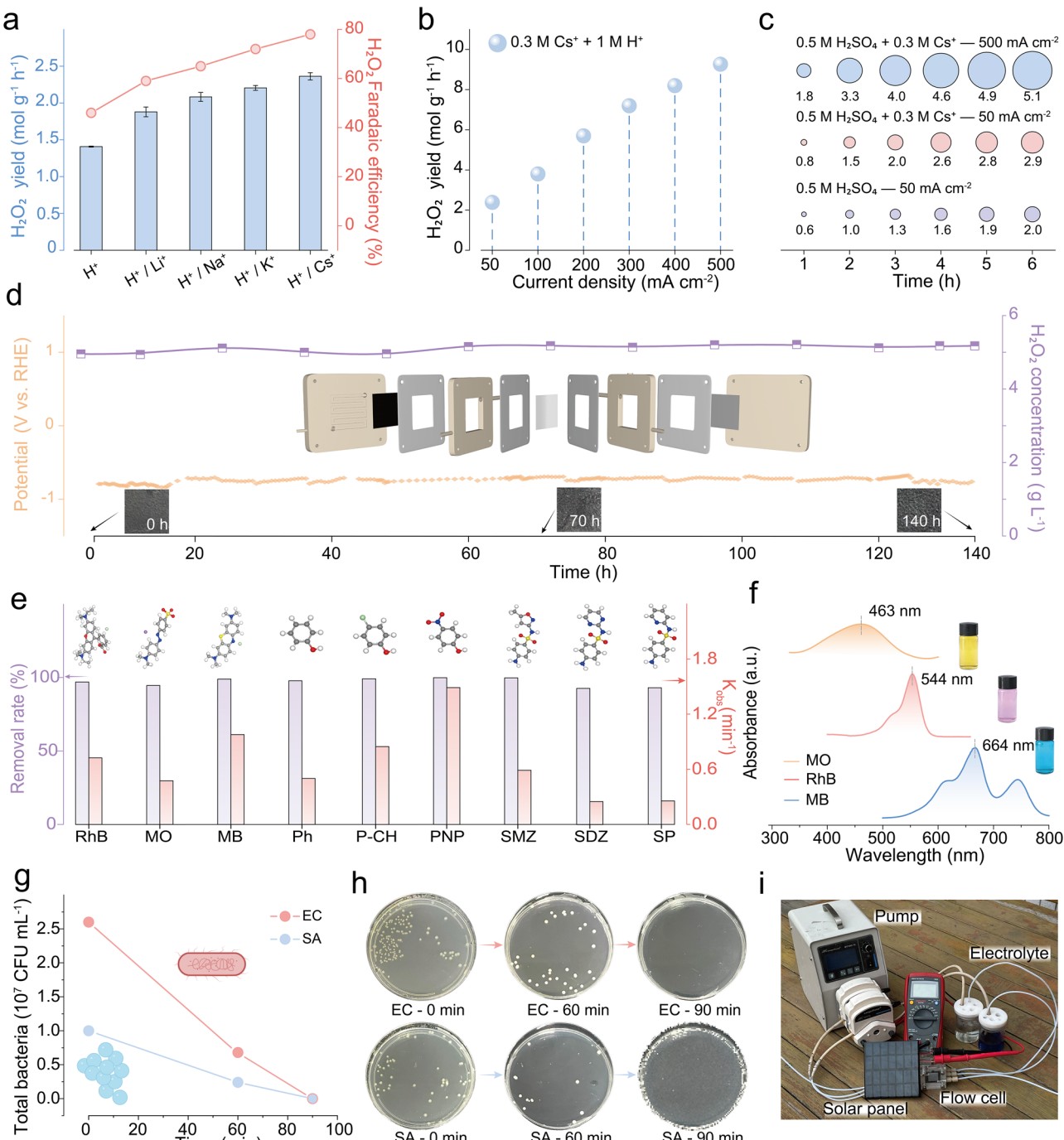

**Fig. 6 | On-site $H_2O_2$ generation in flow cells and related applications. a** $H_2O_2$ production yield of Co-CNT in the $O_2$-saturated 0.5 M $H_2SO_4$ and 0.5 M $H_2SO_4$ + 0.15 M $A_2SO_4$ (A = Li, Na, K, or Cs) electrolyte. **b** $H_2O_2$ yield rate (mol $g_{cat}^{-1}$ $h^{-1}$) as a function of current density, showing a progressive increase culminating in a maximum at the industrially relevant 500 mA $cm^{-1}$ (approximately four-fold higher than at 50 mA $cm^{-1}$) **c** The 6 h-continuous accumulated $H_2O_2$ amount in the flow cell of Co-CNT catalyst under different electrolytes and current densities. **d** Long-term $H_2O_2$ production stability and cell potential at a constant current density of 50 mA $cm^{-1}$ over 140 h. Electrolyte flow rate = 5 mL $h^{-1}$. The insets illustrate the structure of the flow cell and the changes in the cathode surface over time. **e** Fenton degradation performance test for different types of pollutants. The insets show the molecular models of each pollutant. **f** UV-vis absorption spectra of Methyl Orange (MO), Methylene Blue (MB), and Rhodamine B (RhB). The insets show the three dye solutions. **g** Change in bacterial colony-forming units (CFU) and **h** corresponding photographic records of colonies after treatment with $H_2O_2$ produced over 0, 60, and 90 min of electrolysis. The insets in (**h**) show the cells of two types of bacteria. **i** Photograph showing a solar panel-driven flow cell electrolyzer for on-site $H_2O_2$ production toward dye removal. Data is presented as mean ± s.d. ($n \geq 3$). Source data for Fig. 6 are provided as a Source Data file.

cooled mercury cadmium telluride (MCT) detector. A conventional three-electrode configuration was employed for electrochemical control1. In situ SEIRAS measurements were conducted in electrolytes with pH 0, comprising 0.5 M $H_2SO_4$ (Sigma-Aldrich, 70 wt%, 99.999% trace metals basis) and 0.1–0.5 M sulfate solutions containing $Li^+$, $Na^+$, $K^+$, or $Cs^+$ ($Li_2SO_4$, Sigma-Aldrich, 99.99%; $Na_2SO_4$, Sigma-Aldrich, 99.9%; $K_2SO_4$, Sigma-Aldrich, 99.99%; $Cs_2SO_4$, Sigma-Aldrich, 99.9%). A platinum foil and an Ag /AgCl electrode served as the counter

electrode and reference electrode, respectively. Unpolarized infrared radiation was focused onto the interface at an incident angle of 60° using an external reflection accessory (Elema), and the reflected radiation was detected. Prior to ATR-SEIRAS experiments, the gold-coated Si prism surface was electrochemically cleaned by cycling the potential between 0.05 V and 1.1 V vs. the reversible hydrogen electrode (RHE). Time-resolved spectra were recorded within the working electrode potential window at a resolution of 8 cm⁻¹, co-adding 44 scans per spectrum. The reference spectrum was collected at 1.1 V vs. RHE[69].

## Electrochemical measurements

Prior to each measurement campaign, the Ag /AgCl reference electrode was calibrated against a freshly prepared saturated calomel electrode (SCE, 0.241 V vs. SHE at 25 °C) to ensure potential accuracy and reproducibility. Both electrodes were immersed in a thermostatted 3.0 M KCl solution maintained at $25.0 \pm 0.1\,°C$ using a water circulator. The open-circuit potential difference was measured using a high-impedance voltmeter after stabilization for 10 min, with data recorded at 1 s intervals over a 5 min period. The potential of the Ag /AgCl electrode versus the standard hydrogen electrode (SHE) was calculated as $E_{Ag/AgCl\ vs.\ SHE} = E_{SCE\ vs.\ SHE} + \Delta E$, where $\Delta E$ is the measured potential of the Ag /AgCl electrode relative to the SCE. Only electrodes exhibiting a potential within ±2 mV of the theoretical value (e.g., +0.197 V vs. SHE for saturated KCl internal filling) and drift less than 0.5 mV over the measurement period were deemed acceptable for subsequent electrochemical experiments. All potentials reported herein have been corrected to the SHE scale based on this calibration.

The apparent radius of the microelectrode was determined using the steady-state limiting current ($I_{lim}$) according to the following equation:

$$I_{lim} = 4nFDCr \tag{1}$$

where $n$ represents the number of electrons transferred per molecule, $F$ is the Faraday constant (96485 C mol⁻¹), $D$ denotes the diffusion coefficient of the reactant (cm² s⁻¹), $C$ is the bulk concentration of the reactant (mol cm⁻³), and $r$ signifies the true geometric radius of the microdisk electrode. Specifically, $I_{lim}$ was measured as the steady-state limiting current for the reduction of 10 mM [Ru(NH₃)₆]Cl₃ in a 0.1 M K₂SO₄ solution. For this well-established redox couple, the parameters are defined as follows: $n = 1$ (one-electron transfer), $F = 96,485$ C mol⁻¹, $D = 5.71 \times 10^{-6}$ cm² s⁻¹ (diffusion coefficient of [Ru(NH₃)₆]³⁺), and $C = 10$ mM (bulk concentration of [Ru(NH₃)₆]Cl₃). All electrochemical measurements were conducted at ambient temperature ($25 \pm 1\,°C$) using a CHI 760E electrochemical workstation.

In a custom three-electrode system, linear sweep voltammograms of the Au microelectrode were obtained by performing hydrogen evolution experiments in 0.5 M H₂SO₄ solutions containing $x$ M A⁺ (where $x = 0.1$–0.5; A⁺ = Li⁺, Na⁺, K⁺, Cs⁺) at a scan rate of 0.02 V s⁻¹.

Rotating disk electrode (RDE) tests were conducted using a Pine Research Instrumentation MSR rotator coupled to a CHI-760D bipotentiostat. Polarization curves were recorded on a polished Au RDE (5 mm diameter) in hydrogen-saturated 0.1 M H₂SO₄ + 0.3 M K⁺ solution at 0.02 V s⁻¹.

Cyclic voltammograms for [Ru(NH₃)₆]Cl₃ reduction were acquired in a custom three-electrode system using aqueous solutions containing 10 mM [Ru(NH₃)₆]Cl₃ and x M K⁺ (x = 0.1–0.3). Scans were performed from 0.7 V to −0.2 V_SHE at 0.02 V s⁻¹ to determine the limiting current.

Electrocatalytic ORR performance was evaluated using a CHI 760e electrochemical workstation (CH Instruments, Inc., Shanghai, China) with a standard three-electrode configuration and an RRDE (Pine Instruments). The system comprised an Ag /AgCl reference electrode, Pt foil counter electrode, and an RRDE working electrode with a glassy carbon disk (diameter: 5.61 mm, area: 0.247 cm²) and Pt ring (inner diameter: 6.25 mm, outer diameter: 7.92 mm, area: 0.186 cm²). All electrodes were mechanically polished and rinsed with ultrapure water prior to measurements.

Catalyst ink was prepared by dispersing 5 mg synthesized catalyst in a mixture of 480 μL isopropanol, 480 μL ultrapure water, and 20 μL (5.0 wt%) Nafion solution, followed by 2 h sonication. A 10 μL aliquot of the homogeneous ink was drop-cast onto the disk electrode to achieve a catalyst loading of 0.2 mg cm⁻². All electrochemical potentials were converted to the reversible hydrogen electrode (RHE) scale using:

$$E(RHE) = E(Ag/AgCl) + 0.0592 \times pH + 0.197 \tag{2}$$

All electrochemical tests were conducted at room temperature (20 °C) in 0.5 M H₂SO₄ electrolyte thoroughly saturated with N₂ or O₂ prior to measurements. Before electrochemical testing, all catalysts were electrochemically activated via cyclic voltammetry (CV) at 50 mV s⁻¹ until stable voltammograms were obtained. Linear sweep voltammetry (LSV) was performed under O₂-saturated conditions at a scan rate of 10 mV s⁻¹ with an RRDE rotation speed of 1600 rpm, while the ring electrode potential was held constant at 1.2 V_RHE to monitor H₂O₂ production. H₂O₂ selectivity and electron transfer number (n) for the RRDE were calculated using the following equations:

$$Selectivity(\%) = 200 \times (Ir/N)/(Id + Ir/N) \tag{3}$$

$$Electron\ transfer\ number(n) = 4 \times Id/(Id + Ir/N) \tag{4}$$

where $Ir$ and $Id$ denote the ring and disk currents, respectively, and $N$ represents the collection efficiency of the ring electrode (0.37). Catalyst stability was evaluated via chronoamperometry at a constant potential of 1.2 V_RHE.

To probe active sites, poisoning experiments were conducted by adding 10 mM KSCN to the 0.5 M H₂SO₄ electrolyte. Linear sweep voltammetry (LSV) was performed under identical configurations as described above. For hydrogen peroxide reduction reaction (H₂O₂RR) tests, electrolytes comprised N₂-saturated 0.5 M H₂SO₄ containing 10–50 mM H₂O₂ (10, 20, 30, 40, 50 mM).

Assuming metallic Co serves as the active site and all metal sites participate in the reaction, the TOF is calculated as:

$$TOF = IM_{metal}/n\,Fm_{catalyst}\,w_{metal} \tag{5}$$

The current density $I$ (unit: A) for H₂O₂ generation can be obtained by multiplying the disk current by the Faraday efficiency of H₂O₂. $M_{metal}$ (unit: g/mol) represents the molar mass of the transition metal. The parameter $n = 2$ denotes the number of electrons transferred during the H₂O₂ production process. $F$ stands for the Faraday constant, while $m_{catalyst}$ (unit: g) indicates the mass of the catalyst loaded on the electrode. The mass fraction of metal in the catalyst, $w_{metal}$, is determined through ICP analysis. This approach, which presumed all Co atoms in the clusters are active, yielded an overestimated count of the apparent active site. This, in turn, resulted in a calculated TOF that systematically underestimated the actual catalytic activity.

The Tafel slope is calculated by the Tafel equation:

$$\eta = b\log(j/j_0) \tag{6}$$

$\eta$ represents overpotential, $b$ is the Tafel slope, $j$ is the current density, and $j_0$ is the exchange current density.

## H₂O₂ concentration quantification

H₂O₂ content was determined via potassium titanyl oxalate (K₂TiO(C₂O₄)₂) titration. In this method, titanium ions react with H₂O₂

under acidic conditions to form a stable orange peroxotitanium complex, with color intensity proportional to $H_2O_2$ concentration. Experimentally, $H_2O_2$ solutions of known concentration were mixed with a solution containing 3 M $H_2SO_4$ and 0.1 M potassium titanyl oxalate. The absorbance of the mixture was measured at 400 nm using UV-Vis spectroscopy (20 ˚C). A standard calibration curve correlating $H_2O_2$ concentration ($C$) and absorbance ($a$) was established:

$$a = 0.2028C + 0.0101 (R^2 = 0.9991) \tag{7}$$

The Faradaic efficiency (FE) of $H_2O_2$ generation was calculated via the following equation:

$$FE = 2FCV/It \tag{8}$$

where $F$ represents the Faraday constant (96485 C mol$^{-1}$), $C$ refers to the $H_2O_2$ concentration (mol L$^{-1}$) in the electrolyte, $V$ indicates the volume of electrolyte (L) in the cell, and $Q$ is the total charge amount (C) consumed. $I$ represents the current, and $t$ refers to the running time. $Q$ was determined by the integration area of $I$ over $t$ in this chronoamperometry experiment.

## Flow cell $H_2O_2$ production

Practical $H_2O_2$ electrosynthesis performance was evaluated in a two-compartment flow cell with a three-electrode configuration. An Ag /AgCl electrode served as the reference electrode and a Pt foil as the counter electrode. Catalyst ink was uniformly drop-cast onto a $2 \times 2$ cm$^2$ carbon paper to form a catalyst layer, creating a gas diffusion electrode (GDE) with a loading of 0.1 mg·cm$^{-2}$. The GDE was assembled as the oxygen reduction reaction (ORR) cathode in a three-phase flow cell: the catalyst layer faced the cathode chamber while the opposite side was exposed to $O_2$ flow. The cell comprised a proton exchange membrane (Nafion 117, size: 2.5 cm $\times$ 2.5 cm, thickness: 183 μm)-separated cathodic and anodic compartments ($2 \times 4$ mL), with a Pt plate anode and an Ag /AgCl reference electrode fixed in the cathode chamber. The proton exchange membrane was not subjected to any pretreatment.

The entire experimental process was carried out at room temperature (20 °C). The corresponding electrolyte was prepared by adding a certain amount of solid salt to the sulfuric acid solution. Note that a fresh electrolyte was prepared 30 min prior to each test.

## Fenton reaction for pollutant degradation

In situ-generated $H_2O_2$ was coupled with the classical Fenton reaction to achieve efficient pollutant degradation. The reaction scheme is expressed as:

$$2Fe^{2+} + H_2O_2 \rightarrow 2Fe^{3+} + OH^- + \cdot OH \tag{9}$$

This reaction was implemented in the flow cell by introducing both the pollutant solution and $Fe^{2+}$ into the catholyte compartment. Through this process, hydroxyl radicals (·OH) generated from the reaction between in situ-produced $H_2O_2$ and $Fe^{2+}$ continuously degrade pollutants in the solution, enabling efficient contaminant removal.

## MD simulations

The MD simulations were carried out using GROMACS 2020.3 software package with the OPLS-AA force fields[70–72]. To simulate the experiment solutions, 20 hydronium ions, 10 sulfate anions, and around 1080 water molecules were inserted randomly into cubic boxes with side length of 32 Å for 0.5 M $H_2SO_4$ without AMCs. For 0.5 M $H_2SO_4$ with 0.15 M $A_2SO_4$ (A = Li, Na, K, or Cs), 6 AMCs and 3 more sulfate anions were added into the boxes. The steepest descent method was then used to perform 1000 steps of minimization. The models were pre-

equilibrated by NVT ensemble simulations for 200 ps. This was followed by extensive equilibration by 2 ns NPT and 2 ns NVT. The timestep was 1 fs in all simulations. The Berendsen method was used to control both temperatures and pressures at 298 K and 1.0 atm[73]. The Columbic integrations were treated with the particle mesh Ewald (PME) method[74]. These models were used to analysis the number of HBs and the RDFs of AMCs verse water[39,75–79].

To study the connectivity of HB networks, we also built cubic boxes (7 Å in length) filled with 50 water molecules, 2 hydronium ions and a sulfate anion. A hydronium ion was replaced by an AMC for solutions with AMCs. Same relaxations as described above were applied to these systems. PIMD simulations were then conducted using built-in PINT module in CP2K software package[80]. 8 beads are adopted to sample the quantum nuclei. The PIMD trajectories were used for HB networks analyses by ChemNetworks software[81]. An O···H bond distance cutoff of 2.5 Å and a bond angle cutoff of 145° were employed.

## DFT calculations

The free energy profiles of the ORR pathway were investigated by DFT methods using the Vienna ab initio simulation package (VASP) version 5.4.4[82–85]. The core−valence electron interaction was described by the projector augmented wave (PAW) approach with a plane-wave cutoff energy of 400 eV[86]. The electronic exchange−correlation effect was represented by the Perdew−Burke−Ernzerhof (PBE) functional within the generalized gradient approximation (GGA)[87,88]. The Grimme D3 correction was added to describe the London dispersion integrations[89]. All calculations are spin-polarized. The entropy correction and the zero-point energy correction at 298.15 K were taken into consideration for the Gibbs free energy. The 2e$^-$ reaction pathway for $H_2O_2$ production is calculated as following:

$$O_2(g) + 2H^+ + 2e^- + * \rightarrow *OOH + H^+ + e^- \tag{10}$$

$$*OOH + H^+ + e^- \rightarrow H_2O_2(g) + * \tag{11}$$

where * represents the catalyst, and *OOH donates the intermediate species adsorbed on the catalyst. For the Gibbs free energy, both entropy and the zero-point energy correction were considered at 298.15 K. The Gibbs free energy differences were calculated:

$$\Delta G_{(1)} = \Delta G_{OOH} - -4.92 (eV) \tag{12}$$

$$\Delta G_{(2)} = 3.56 - -\Delta G_{OOH} (eV) \tag{13}$$

where $\Delta G_{OOH}$ represents the adsorption energy of molecular OOH.

We applied a constant potential (+0.15 V $vs$. SHE, consistent with experiments) to the ORR simulations by CP-VASP[90,91]. The targeted Fermi level referenced to the electrolyte was set to −4.75 V with a convergence criterion of 0.01 V for the electrolyte-referenced Fermi level. The solvation model during in calculations was considered by using the VASPsol++ framework implemented in VASP[92].

Each AMC was put on the surface of Co-CNT and the adsorption energies were calculated by DFT:

$$E_{ad} = E_{total} - E_{Co-CNT} - E_{AMC} \tag{14}$$

where $E_{total}$ is the total energy of an AMC adsorbed on the catalyst, while $E_{Co-CNT}$ and $E_{AMC}$ represent the energies of Co-CNT and AMC, separately. Positive-charged calculations were performed by manually setting the number of valence electrons to consider the positive charge state of AMCs.

To study the proton transfer process, we constructed 3 clusters of 8, 8 and 10 water molecules with an extra proton, for 1-HB to 4-HB,

2-HB to 4-HB and 4-HB to 4-HB, respectively. Flexible scans were performed for the O···H···O bond between the proton donor and the proton acceptor using Jaguar program at the BLYP/6-31 G* level[93]. The proton transfer energy barrier was calculated as the energy difference between the intermediate state with highest energy and the initial structure. All the files of optimized computational models are provided in Supplementary Data 1.

Due to the inherent limitations of simulations in capturing the full complexity of real catalytic environments, these results serve as theoretical references requiring experimental validation.

## Data availability
The data supporting the findings of this study are available within the article and its Supplementary Information files. Source data are provided with this paper.

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

## Acknowledgments

This work was supported by the was financially supported by the National Natural Science Foundation of China (52572251, J.Y.S., 92472110, T.C., and 22173066, T.C.). Incubation Foundation of Beijing University of Technology (PY202101, Y.F.W.) and the Natural Science Foundation of Jiangsu Province (BK20230065, T.C.), This work was partly supported by Collaborative Innovation Center of Suzhou Nano Science & Technology.

## Author contributions

Y.F.W. and J.Y.S. conceived the project. Y.F.W., H.Y., T.C., and J.Y.S. supervised the project. P.Y.D., H.W., B.B.L., and H.Y.Z. carried out the experimental tests. Y.Q.L., T.C., and H.Y. performed theoretical calculations. Y.F.W., P.Y.D., Y.Q.L., H.W., B.B.L., H.Y.Z., and H.Y. discussed the results. P.Y.D., Y.Q.L., H.Y., T.C., Y.F.W., and J.Y.S. analyzed the core data and wrote the manuscript. All authors agreed on the manuscript and approved the submission.

## Competing interests

The authors declare no competing interests.
