## [Transparent Peer Review file · Nature Communications]

Differentiating interfacial water structures via alkali metal cation promotor for H₂O₂ electrosynthesis in acid

Corresponding Author: Professor Jingyu Sun

Version 0:

Reviewer comments:

Reviewer #1

(Remarks to the Author)

The study investigates how alkali metal cations (AMCs: Li⁺, Na⁺, K⁺, Cs⁺) enhance H₂O₂ production during the oxygen reduction reaction (ORR). The authors combine a wide range of experimental and computational techniques, including physicochemical analyses (XRD, TEM, ICP, XPS, FT-IR), electrochemical characterization, in situ ATR-SEIRAS, MD and DFT simulations, as well as a flow-cell demonstration coupled with a Fenton reaction for pollutant degradation.

While the conventional interpretation would be that larger cations simply block or hinder the 4-electron ORR pathway at the active sites or the outer Helmholtz plane, the authors propose a more sophisticated mechanism; water network, Eigen-Zundel interconversion, and proton hopping. The results provide a coherent and convincing narrative that the altered interfacial water structure governs the ion-specific enhancement of H₂O₂ selectivity. The mechanistic consistency across experiments and simulations is a notable strength of the work, but there are several comments to further develop the research.

Comments:

1. It would be beneficial to include evidence (e.g., from FT-IR, XPS, or DFT adsorption-energy calculations) clarifying whether the introduced AMCs interact with or adsorb onto the catalyst surface, or remain fully hydrated near the interface.
2. The MD and DFT models appear to describe neutral interfaces. In reality, the electrochemical reactions occur under an applied potential, especially in the presence of charged species like Cs⁺. A discussion of possible approaches—or the limitations of the current setup—to model potential-controlled interfaces would strengthen the computational section.
3. The flow-cell experiment successfully illustrates the practical applicability of H₂O₂ generation, but it contributes little to the mechanistic understanding, which is the core focus of this paper. This section could be presented more clearly as an application-oriented addendum rather than mechanistic evidence.

Reviewer #2

(Remarks to the Author)

I have read the manuscript "Differentiating interfacial water structures via alkali metal cation 1 promotor for H₂O₂ electrosynthesis in acid". Overall, the manuscript presents interesting findings and is addressing the relevant topic. However, there are some issues I feel need to be addressed before publication

1. "Employing Co-CNT as a model catalyst, we observe that the 2e⁻ ORR selectivity in acid follows a monotonically increasing trend from to which is not revealed by prior studies (Figure 1)."

And similar claims authors make throughout the manuscript, while strictly speaking true for this specific catalyst, are not generally applicable. The exact same cation trend for the reaction has been recently observed, e.g., for facets of graphitic carbon <https://doi.org/10.1021/acscatal.4c04734>

Additionally, I think the authors should comment on this and similar literature findings.

In fact, I find that one of the biggest weaknesses of the manuscript currently is the very limited referral to literature data for oxygen reduction to hydrogen peroxide. For instance, Metal-CNT (<https://doi.org/10.1021/cr900136g>, <https://doi.org/10.1002/anie.201509241>, <https://doi.org/10.1016/j.jcat.2005.11.024>), discussions about 2-electron ORR (<https://doi.org/10.1021/acscatal.8b00217>), and cation (<https://doi.org/10.1021/acscatal.4c04734>) and electrolyte effects in

general (

<https://doi.org/10.1002/cssc.200800257>

, <https://www.sciencedirect.com/science/article/abs/pii/S0021951704001952-33,127,131>[https://doi.org/10.1016/S0926-3373\(02\)00232-1](https://doi.org/10.1016/S0926-3373(02)00232-1)). This is by no means an exhaustive list

2. "The varying influences of 85 different AMCs stem from their distinct capabilities to perturb the H-bond network, 86 associated with their hydration properties and coordination strengths. Impressively, the 87 elucidated mechanism is proven to be universal, as Fe-CNT and Ni-CNT catalysts 88 exhibit analogous AMC-dependent behavior"

Once again, this would be an opportunity to refer to and comment on existing literature data on cation effects on other catalysts.

3. Cs⁺ produced the most pronounced suppression, reaching a maximum factor of 13-fold. It is further noted that there existed a concentration-dependent variation of this suppression effect (Supplementary Fig. 3).

Some parts of CVs appear to be missing in Sup. Fig 3C.

While it is true that Cs⁺ has the strongest and Li⁺ the smallest effect in general. Here we do not see the Cs⁺>K⁺>Na⁺>Li⁺ trend in e.g. 0.1M M2SO4

4. An optimized concentration of 0.3 M was selected, since higher dosages were found to 113 impair H₂O₂ generation within the system (Supplementary Fig. 4).

Why would this be? The authors claim that H⁺ concentration suppression is good for 2-electron ORR selectivity, then why would the trend flip at 0.3M?

5. Why did the authors choose to perform all measurements in sulfuric acid? Sulfuric acid is known to strongly interact with many surfaces, complicating interpretation. Since the authors claim that the observed trend are a result of H⁺ availability specifically, the authors should perform measurements in HClO₄ in addition with same amounts of alkaline perchlorates to observe the cations effects here as well.

6. In Fig 2C, as well as Fig 5b, besides a change in the diffusion limiting current, we can also observe a shift in the onset and halfwave potentials for both reactions. If the activity differences stem purely from H⁺ concentration near the surface, this should primarily cause a difference in the current densities.

In fact, the cation concentrations used in this work are quite high. Have the authors measured the pHs of the electrolytes and have any differences been accounted for in the calculation of the applied potentials? Cation effects have been observed in much more subtle concentrations of 10-50 mM, i.e., two orders of magnitude below what the authors are applying!

7. "The addition of AMCs significantly lowered the onset potential and disk current density as compared to the bare electrolyte scenario, which was likely 268 to originate from hindered proton transfer in impeding the HER or 4e⁻ ORR pathways occurring at the catalyst surface^{4,38}" referring to Figure 5b

In Figure 5b we can see that the ORR onset is, in fact, delayed after the addition of the cations, i.e., the reaction onset happens later in the potential sweep.

8. "Our catalyst showcased an outstanding 280 turnover frequency (TOF) value of 0.51 s⁻¹ at 0.5 VRHE (Fig. 5d), which compared 281 favorably with reported electrocatalysts^{39,40}."

Calculating TOF requires knowledge of the precise amount of active sites on the surface. How was this determined and how was it calculated?

9. "To uncover the influence of different electrolyte additives upon catalyst activity, 285 we performed electrochemical double-layer capacitance measurements. As anticipated, 286 the catalyst exhibited the highest electrochemical active surface area value in the Cs⁺-287 dosed electrolyte, indicative of facilitated mass transport at ORR active sites 288 (Supplementary Fig. 37)^{43,44}"

This is, likely, a misinterpretation. The authors do in fact measure higher capacitance, however, since the cations are participating in the double layer the specific capacitance is likely different in the presence of different cations, while the ECSA is likely the same.

10. The authors measure the generated H₂O₂ by oxidation at 1.2 V at the ring electrode, but later, when they measure H₂O₂ decomposition (also by oxidation) they perform the measurement at 1V. What is the reason for this? Is the trend the same at 1.2V?

11. "The on-site generation of H₂O₂ in our system is highly attractive for targeted 329 applications such as wastewater treatment."

How would such high alkaline cation concentrations (0.3M) be applied in practical wastewater treatment. Seems like this would be arguably a worse or equally bad contamination as the contaminants being removed by H₂O₂ generation.

12. Figure 2A the voltammograms are too small to be assessed. Why are the potential windows for the blank and the other

CVS different?

Caption for Figure 5f is missing.

Reviewer #3

(Remarks to the Author)
File attached.

Version 1:

Reviewer comments:

Reviewer #1

(Remarks to the Author)

The revision has addressed most of my previous concerns and the manuscript is now close to acceptance. I believe the work can be accepted after the authors clarify the following minor points to improve transparency and reproducibility.

1. In the rebuttal, the authors state that adsorption energies of alkali metal cations (AMCs) on the catalyst surface were calculated to exclude direct surface binding. It would be helpful to explicitly clarify whether the positive charge state of the cations (e.g., $\text{Li}^+/\text{Na}^+/\text{K}^+$) was considered in these calculations, and how charge compensation was handled in the computational model (e.g., background charge, counter ions, solvation treatment, etc.), since this may influence adsorption energetics and interfacial electrostatics.

2. The authors performed constant-potential DFT calculations at 0.15 V vs. SHE to account for electrochemical conditions. While this value is described as being consistent with experimental conditions, it would strengthen the manuscript to provide a clearer electrochemical/theoretical rationale for selecting this potential, or to briefly discuss how representative it is within the relevant ORR operating window and whether the conclusions are expected to remain robust over a reasonable potential range.

3. In the Methods section, particularly in the "DFT calculations" subsection near the end of the manuscript, there are currently no references cited for the computational methods and models employed. Please add appropriate methodological references (e.g., DFT setup, electrochemical/solvation model, constant-potential method) to improve completeness and reproducibility.

Reviewer #2

(Remarks to the Author)
Dear Authors, Dear Editors,

I feel like all my remarks have been answered appropriately.

Best regards.

Reviewer #3

(Remarks to the Author)

I am satisfied by the additional work the co-authors have done to elevate their contribute to Nature Comms. level. I recommend it's acceptance.

Version 2:

Reviewer comments:

Reviewer #1

(Remarks to the Author)

I recommend acceptance of the revised manuscript. The authors have adequately addressed my previous comments, and the revisions have improved the clarity, transparency, and reproducibility of the work.

Response to Reviewers' Comments (NCOMMS-25-75578)

Reviewer #1 (Remarks to the Author):

The study investigates how alkali metal cations (AMCs: Li^+ , Na^+ , K^+ , Cs^+) enhance H_2O_2 production during the oxygen reduction reaction (ORR). The authors combine a wide range of experimental and computational techniques, including physicochemical analyses (XRD, TEM, ICP, XPS, FT-IR), electrochemical characterization, in situ ATR-SEIRAS, MD and DFT simulations, as well as a flow-cell demonstration coupled with a Fenton reaction for pollutant degradation.

While the conventional interpretation would be that larger cations simply block or hinder the 4-electron ORR pathway at the active sites or the outer Helmholtz plane, the authors propose a more sophisticated mechanism; water network, Eigen-Zundel interconversion, and proton hopping. The results provide a coherent and convincing narrative that the altered interfacial water structure governs the ion-specific enhancement of H_2O_2 selectivity. The mechanistic consistency across experiments and simulations is a notable strength of the work, but there are several comments to further develop the research.

Response: We are grateful for your positive feedback. All requested revisions based on your instructive comments have been carried out.

Specific comment:

1. It would be beneficial to include evidence (e.g., from FT-IR, XPS, or DFT adsorption-energy calculations) clarifying whether the introduced AMCs interact with or adsorb onto the catalyst surface, or remain fully hydrated near the interface.

Reply 1: Thank you for your valuable advice. To directly address the state of the introduced AMCs, we performed supplementary DFT calculations to evaluate the adsorption energies of anhydrous AMCs on the Co-CNT catalyst surface. The computed adsorption energies for all AMCs (Li^+ , Na^+ , K^+ , Cs^+) are positive, indicating that these cations are not thermodynamically favorable to adsorb onto the catalyst surface. Instead, they remain fully hydrated and reside near the electrode–electrolyte interface, as illustrated in Figure 4a. This finding confirms that the promotional effect of AMCs originates from their hydrated state at the interface, rather than through direct adsorption onto the catalytic sites.

Revisions to Manuscript:

Supplementary Fig. 34 | Optimized structures of anhydrous AMCs on Co-CNT and its adsorption energy values.

2. The MD and DFT models appear to describe neutral interfaces. In reality, the electrochemical reactions occur under an applied potential, especially in the presence of charged species like Cs⁺. A discussion of possible approaches—or the limitations of the current setup—to model potential-controlled interfaces would strengthen the computational section.

Reply 2: Thank you for the constructive comment in improving the quality of our manuscript. We fully agree that the calculations of electrochemical reactions under an applied potential could describe the nature of electrochemical reaction mechanism in a more accurate manner.

In response to your suggestion, we have carried out **additional constant-potential DFT calculations using the CP-VASP method (+0.15 V vs. SHE, consistent with key experimental conditions) to explicitly simulate the electrode potential during the ORR^{1,2}**. These calculations confirm that the high selectivity toward the 2e⁻ pathway could be maintained in the presence of AMCs under a representative applied potential (**Fig. R1**). The results further support our central conclusion that the cation-induced restructuring of the interfacial water network, rather than a change in the fundamental reaction mechanism at the active site, is the dominant promotional effect. The reactions remain a high 2e⁻ ORR selectivity with AMCs.

While our present MD simulations employed a charge-neutral setup to efficiently explore the cation-specific perturbation of the hydrogen-bond network, we acknowledge that explicitly modeling the full electrochemical double layer under bias remains a significant computational challenge. Future work will benefit from more advanced simulation techniques to fully couple the interfacial water structure with the electrostatic field. Nevertheless, the strong consistency between our potential-inclusive DFT results, the potential-dependent in situ spectroscopic data, and the observed electrochemical trends provides a coherent and convincing mechanistic picture.

Fig. R1. Free energy profiles of the ORR pathways on Co-CNT catalysts under 0.15 V vs. SHE.

3. *The flow-cell experiment successfully illustrates the practical applicability of H₂O₂ generation, but it contributes little to the mechanistic understanding, which is the core focus of this paper. This section could be presented more clearly as an application-oriented addendum rather than mechanistic evidence.*

Reply 3: We sincerely thank you for raising this important point regarding the role of flow-cell experiments in the context of our manuscript. We fully agree that the flow-cell section primarily illustrates the **practical applicability** of the H₂O₂ generation strategy. In revising the manuscript, we have refined the presentation of this part to more clearly distinguish between **mechanistic investigation** and **application demonstration**, as suggested.

At the same time, the flow-cell performance data (Figure 6) demonstrates that the Cs⁺-promoted system exhibits not only the highest H₂O₂ production rate (9.2 mol g⁻¹ h⁻¹ at 500 mA cm⁻²) but also exceptional operational stability over 140 h, which directly reflects the robustness of the interfacial water restructuring mechanism proposed in our work.

The observed cation-dependent performance trend (Li⁺ < Na⁺ < K⁺ < Cs⁺) in the flow-cell mirrors the trend obtained from fundamental microelectrode, spectroscopic, and computational analyses, thereby **corroborating the consistency of the mechanism under industrially relevant conditions**.

To better frame this section, we have revised the sub-heading to “**H₂O₂ generation and utilization enabled by AMCs**” which more accurately presents the flow-cell experiments as an application-oriented extension that concurrently verifies the practical implications of the proposed mechanism. This format aligns with similar high-impact studies that include flow-cell demonstrations as a logical endpoint of mechanistic discovery^{3,4}.

Revisions to Manuscript:

“**H₂O₂ generation and utilization enabled by AMCs**” in Page 12.

Reviewer #2 (Remarks to the Author):

I have read the manuscript “Differentiating interfacial water structures via alkali metal cation I promotor for H₂O₂ electrosynthesis in acid”. Overall, the manuscript presents interesting findings and is addressing the relevant topic. However, there are some issues I feel need to be addressed before publication.

Response: We are grateful for your positive and constructive feedback. All requested revisions based on your professional comments have been carried out.

Specific comment:

1. “Employing Co-CNT as a model catalyst, we observe that the 2e⁻ ORR selectivity in acid follows a monotonically increasing trend from to which is not revealed by prior studies (Figure 1).”

And similar claims authors make throughout the manuscript, while strictly speaking true for this specific catalyst, are not generally applicable. The exact same cation trend for the reaction has been recently observed, e.g., for facets of graphitic carbon <https://doi.org/10.1021/acscatal.4c04734> Additionally, I think the authors should comment on this and similar literature findings.

In fact, I find that one of the biggest weaknesses of the manuscript currently is the very limited referral to literature data for oxygen reduction to hydrogen peroxide. For instance, Metal-CNT (<https://doi.org/10.1021/cr900136g>, <https://doi.org/10.1002/anie.201509241>, <https://doi.org/10.1016/j.jcat.2005.11.024>), discussions about 2-electron ORR (<https://doi.org/10.1021/acscatal.8b00217>), and cation (<https://doi.org/10.1021/acscatal.4c04734>) and electrolyte effects in general (

<https://doi.org/10.1002/cssc.200800257>, <https://www.sciencedirect.com/science/article/abs/pii/S0021951704001952-33,127,131> [https://doi.org/10.1016/S0926-3373\(02\)00232-1](https://doi.org/10.1016/S0926-3373(02)00232-1)). This is by no means an exhaustive list

Reply 1: Thank you for this critical comment. We fully agree that a comprehensive discussion of the existing literature is essential to enhance the quality of our work. Based on your suggestions and further analysis, we have expanded the discussion to acknowledge and refer to the important prior studies, including cation effect on acidic 2e⁻ ORR, the employment of different catalysts, as well as cation and electrolyte effects in general. The relating revision has been made in the rephrased **Introduction** section, with the supplementary incorporation of advised references (*New Ref. [3–8][30][34]*).

Revisions to Manuscript:

“Hydrogen peroxide (H₂O₂) has emerged as a green and versatile oxidant with vast application

potentials in wastewater treatment, organic synthesis and chemical technology¹⁻⁷. Its on-site electrosynthesis via the two-electron oxygen reduction reaction ($2e^-$ ORR) with enriched choices of electrocatalysts⁸.

Electrolyte engineering, particularly the introduction of alkali metal cations (AMCs: Li^+ , Na^+ , K^+ , or Cs^+)²⁸⁻³⁰, has emerged as a promising strategy to steer interfacial chemistry toward H_2O_2 electrosynthesis in acid^{31,32}. A recent study indicated a graphitic carbon electrode exhibited an increase in $2e^-$ ORR selectivity following the trend of $Cs^+ > K^+ > Na^+ > Li^+$, where different cations would interfere in the $*OOH$ binding and influence the local electric field³⁴” in Page 3-4.

Added references:

- 3 Brillas, E. *et al.* Electro-Fenton Process and Related Electrochemical Technologies Based on Fenton’s Reaction Chemistry. *Chem. Rev.* **109**, 6570-6631 (2009).
- 4 Chinta, S. *et al.* A mechanistic study of H_2O_2 and H_2O formation from H_2 and O_2 catalyzed by palladium in an aqueous medium. *J. Catal.* **225**, 249-255 (2004).
- 5 Ntainjua N., E. *et al.* Effect of Halide and Acid Additives on the Direct Synthesis of Hydrogen Peroxide using Supported Gold–Palladium Catalysts. *ChemSusChem* **2**, 575-580 (2009).
- 6 Choudhary, V. R. *et al.* Role of chloride or bromide anions and protons for promoting the selective oxidation of H_2 by O_2 to H_2O_2 over supported Pd catalysts in an aqueous medium. *J. Catal.* **238**, 28-38 (2006).
- 7 Burch, R. *et al.* An investigation of alternative catalytic approaches for the direct synthesis of hydrogen peroxide from hydrogen and oxygen. *Appl. Catal., B* **42**, 203-211 (2003).
- 8 Yang, S. *et al.* Single-Atom Catalyst of Platinum Supported on Titanium Nitride for Selective Electrochemical Reactions. *Angew. Chem. Int. Ed.* **55**, 2058-2062 (2016).
- 27 Shirley, J. C. *et al.* Reevaluating Anomalous Electric Fields at the Air-Water Interface: A Surface-Specific Spectroscopic Survey. *J. Chem. Phys.* eprint arXiv:2508.15422 (2025).
- 28 Hübner, J. L. *et al.* Cation Effects on the Acidic Oxygen Reduction Reaction at Carbon Surfaces. *ACS Energy Lett.* **9**, 1331-1338 (2024).
- 30 Yang, S. *et al.* Toward the Decentralized Electrochemical Production of H_2O_2 : A Focus on the Catalysis. *ACS Catal.* **8**, 4064-4081 (2018).
- 34 Olean-Oliveira, A. *et al.* Electrochemical Insights into Hydrogen Peroxide Generation on Carbon Electrodes: Influence of Defects, Oxygen Functional Groups, and Alkali Metals in the Electrolyte. *ACS Catal.* **14**, 17675-17689 (2024).

2. “The varying influences of different AMCs stem from their distinct capabilities to perturb the H-bond network, associated with their hydration properties and coordination strengths. Impressively, the elucidated mechanism is proven to be universal, as Fe-CNT and Ni-CNT catalysts exhibit

analogous AMC-dependent behavior”

Once again, this would be an opportunity to refer to and comment on existing literature data on cation effects on other catalysts.

Reply 2: Thanks for your kind suggestion. In the revised **Introduction** section, we have expanded our discussion on cation effects using other catalysts based on existing literatures. Our claim of “universality” is carefully confined in the catalyst type of our employed metal-CNT, since it has not been reported in the cation-sensitive acidic $2e^-$ ORR study. By integrating these discussions, we aim to present our work not as an isolated finding, but as a conceptual bridge that connects electrolyte engineering, interfacial water science and catalyst design for sustainable H_2O_2 production.

Revisions to Manuscript:

“Particularly using metal-free carbon-based electrocatalysts, Zhang et al. observed that H_2O_2 production rates can be improved by an AMC-induced shielding effect, yet were relatively unaffected by the AMC types³³. A recent study indicated a graphitic carbon electrode exhibited an increase in $2e^-$ ORR selectivity following the trend of $Cs^+ > K^+ > Na^+ > Li^+$, where different cations would interfere in the $*OOH$ binding and influence the local electric field³⁴. It is interesting to note that the AMC roles in acidic $2e^-$ ORR manifest inconsistent trends. In further contexts, the cation effect is effective across other catalyst types apart from carbonaceous materials. For instance, Cao *et al.* found the H_2O_2 production followed the trend of $K^+ > Na^+ > Li^+ \sim Cs^+$ with the employment of earth-abundant TiC catalyst, which was dictated by the hydrated ionic radius of AMCs³⁵. These disparities imply that the dominant mechanism might be sensitive to specific conditions, such as catalyst identity, cation concentration and operating current density. A coherent picture of how AMCs reshape the interfacial environment in acidic $2e^-$ ORR is still lacking, especially in the realm of prevailing transition metal-involved carbon electrocatalysts (e.g., metal-carbon nanotube, M-CNT).” in Page 3-4.

Added references:

34 Olean-Oliveira, A. *et al.* Electrochemical Insights into Hydrogen Peroxide Generation on Carbon Electrodes: Influence of Defects, Oxygen Functional Groups, and Alkali Metals in the Electrolyte. *ACS Catal.* **14**, 17675-17689 (2024).

3. Cs^+ produced the most pronounced suppression, reaching a maximum factor of 13-fold. It is further noted that there existed a concentration-dependent variation of this suppression effect (Supplementary Fig. 3).

Some parts of CVs appear to be missing in Sup. Fig 3C.

While it is true that Cs^+ has the strongest and Li^+ the smallest effect in general. Here we do not see

the $Cs^+ > K^+ > Na^+ > Li^+$ trend in e.g. 0.1M M_2SO_4 .

Reply 3: Thanks for your meticulous review and valuable comments. **Following these suggestions, we have re-examined the data.** Firstly, the complete plateau region (from -0.4 to -1.1 V) of the LSV profiles for certain conditions (especially at lower cation concentrations) has been presented in *New Supplementary Fig. 4*. Secondly, the AMC concentration-dependent variation of diffusion-limiting current density has been updated in *New Supplementary Fig. 5*. In this sense, it could be observed that the dosed AMC with varied concentrations (0.1–0.4 M; *i.e.*, including 0.1 M) all exhibits suppression effect ($Li^+ < Na^+ < K^+ < Cs^+$) on proton transport. The strength of suppression scales monotonically with both cation type and concentration, reinforcing the proposed mechanism that larger cations (Cs^+) could more effectively reconstruct the interfacial water and impede proton hopping via H-bond network.

Revisions to Manuscript:

“It is further noted that there existed a concentration-dependent variation of this suppression effect (Supplementary Figs. 4 and 5).” in Page 5.

Supplementary Fig. 4 | HER polarization curves of Au microelectrodes in 0.5 M $H_2SO_4 + x$ M $Li^+/Na^+/K^+/Cs^+$ (x ranging from 0.1 to 0.4) at 0.02 V s^{-1} .

Supplementary Fig. 5 | Diffusion-limited current density for 1 M H⁺ and 1 M H⁺ + x M A⁺ (x ranging from 0.1 to 0.4; A = Li, Na, K, or Cs).

4. An optimized concentration of 0.3 M was selected, since higher dosages were found to impair H₂O₂ generation within the system (Supplementary Fig. 4).

Why would this be? The authors claim that H⁺ concentration suppression is good for 2-electron ORR selectivity, then why would the trend flip at 0.3M?

Reply 4: Thank you for the professional suggestion on the experimentally optimized concentration at 0.3 M and the subsequent decline in H₂O₂ production performances at higher concentrations. The question refers to a key interfacial reaction engineering: the balance between suppressing **parasitic pathways and maintaining sufficient reaction kinetics**.

Our work demonstrates that the proton-delivery suppression by AMCs boosts 2e⁻ ORR activity. However, this does not imply that proton availability should be minimized indefinitely. The overall ORR rate, including the desired 2e⁻ pathway, remains a proton-coupled electron transfer process. Excessively hindering the proton transport would hence slow down the entire reaction, leading to the observed decline in H₂O₂ production rate at quite high cation concentrations. The optimum at ~0.3 M reflects **a trade-off between selectivity and activity**. Below this concentration, proton availability remains too high, favoring 4e⁻ ORR and H₂O₂ reduction. Above this concentration, proton supply becomes kinetically limiting, reducing the overall ORR current and hence the H₂O₂ generation rate.

It is worth-noting that such an optimized concentration alters systematically with the cation identity: it occurs at ~0.3 M for Cs⁺, ~0.3 M for K⁺, ~0.4 M for Na⁺, and ~0.5 M for Li⁺ (*New Supplementary Fig. 6*). This trend aligns precisely with each typed cation's capability to perturb the interfacial H-bond network and suppress proton transport (Cs⁺ > K⁺ > Na⁺ > Li⁺). A stronger “suppressor”

requires a lower concentration to reach the same level of proton limitation, explaining why the peak performance occurs at lower concentrations for Cs⁺ than for Li⁺.

More significantly, from an **environmental engineering perspective**, this concentration-dependent behavior is meaningful. Industrial wastewater streams often contain fluctuating ionic strength; our survey indicate that even at moderate cation concentrations (≥ 0.2 M), the promoting effect remains robust and follows a predictable order. This provides a practical guideline for electrolyte management in real applications, such as using brine or high-salinity wastewater as an electrolyte source for on-site H₂O₂ electrosynthesis.

Revisions to Manuscript:

“The optimized AMC concentration refers to a key interfacial reaction engineering of the balance between suppressing parasitic pathways and maintaining sufficient reaction kinetics. Considering the trade-off between proton availability and reaction kinetics, a medium concentration of 0.3 M was selected to assess the AMC concentration effect on the H₂O₂ yield (Supplementary Fig. 6).” in Page 5.

Supplementary Fig. 6 | H₂O₂ yield based on Co-CNT at different AMC concentrations in a flow-cell reactor.

5. *Why did the authors choose to perform all measurements in sulfuric acid? Sulfuric acid is known to strongly interact with many surfaces, complicating interpretation. Since the authors claim that the observed trends are a result of H⁺ availability specifically, the authors should perform measurements in HClO₄ in addition with same amounts of alkaline perchlorates to observe the cations effects here as well.*

Reply 5: We sincerely thank you for raising this important point. The choice of sulfuric acid as the

primary electrolyte was motivated by its **widespread environmental relevance**. The SO_4^{2-} is among the most abundant anions in industrial wastewater, mining effluent, and natural saline waters, making H_2SO_4 a practically significant medium for evaluating electrochemical processes aimed at real-world water treatment applications.

We fully agree that SO_4^{2-} may exhibit specific adsorption or surface interactions that could potentially complicate the interpretation of cation effects. To rigorously decouple the role of anions and confirm that the observed trends originate primarily from cation-modulated proton availability, given the low solubility of HClO_4 , **we have followed the suggestion and supplemented a complete set of control experiments using HClO_4 with corresponding alkali perchlorates** (0.1 M HClO_4 + 0.05 M AClO_4 , A = Li, Na, K, or Cs). The introduction of AMCs in perchlorate media again enhances the H_2O_2 production. The cation-dependent promotion trend follows the same order: $\text{Cs}^+ > \text{K}^+ > \text{Na}^+ > \text{Li}^+$, in good agreement with the trend observed in SO_4^{2-} -based electrolytes. The relating result is showing below for your kind check (**Fig. R2**).

Fig. R2. The H_2O_2 concentration based on Co-CNT for 0.1 M HClO_4 and 0.1 M HClO_4 + 0.05 M AClO_4 in a flow-cell reactor after 1 h (A = Li, Na, K, or Cs).

6. In Fig 2c, as well as Fig 5b, besides a change in the diffusion limiting current, we can also observe a shift in the onset and halfwave potentials for both reactions. If the activity differences stem purely from H^+ concentration near the surface, this should primarily cause a difference in the current densities.

In fact, the cation concentrations used in this work are quite high. Have the authors measured the pHs of the electrolytes and have any differences been accounted for in the calculation of the applied potentials? Cation effects have been observed in much more subtle concentrations of 10-50 mM, i.e., two orders of magnitude below what the authors are applying!

Reply 6: Thank you for this insightful comment. The shifts in both onset potential and half-wave potential indeed reflect a comprehensive modulation of the ORR process by alkali metal cations (AMCs).

Onset potential: The onset potential (defined here as the potential at which the current density reaches -0.05 mA cm^{-2}) corresponds to the initial activation of O_2 . Its negative shift indicates that the adsorption and activation of O_2 are hindered in the presence of AMCs, consistent with a reduction in proton availability that impedes the formation of the $^*\text{OOH}$ intermediate. The negative shift of the half-wave potential further reflects a slowdown in the overall ORR kinetics, primarily due to the suppression of the $4e^-$ pathway that requires more protons compared to the $2e^-$ pathway, with a lower activation barrier under these conditions, is less affected. This interpretation aligns with the simultaneous decrease in the limiting current density in the LSV curves (Fig. 5b), which signals a shift from the $4e^-$ to the $2e^-$ pathway. The shift in potential in Fig. 2c is also attributed to the suppression of electrode surface reactions caused by AMC.

Cation concentration: We have measured the pH values in varied electrolytes (0.5 M H_2SO_4 with 0.15 M A_2SO_4) and found that the values remained virtually unchanged at $\sim 0.41 (\pm 0.02)$ (New Supplementary Fig. 8). In converting the applied potentials to the RHE scale using $E(\text{RHE}) = E(\text{Ag}/\text{AgCl}) + 0.0592 \times \text{pH} + 0.197$, such minor pH variations introduce a negligible shift ($< 2 \text{ mV}$). Therefore, the observed potential offsets originate principally from cation-induced interfacial water restructuring rather than pH changes.

The higher cation concentrations used here (0.3 M) were aimed to emphasize the restructuring effect on the interfacial H-bond network and its pronounced impact on proton transport under industrially relevant current densities. We have supplemented H_2O_2 production tests at lower AMC concentrations (10 or 50 mM), which confirm that the promotional trend ($\text{Cs}^+ > \text{K}^+ > \text{Na}^+ > \text{Li}^+$) persists even at lower ionic strengths (Fig. R3). This consistency bridges our observations with reported cation effects at much lower concentrations (10–50 mM) and underscores the general role of AMCs in tuning proton delivery through interfacial water reconfiguration.

Revisions to Manuscript:

“To exclude potential interference from other variables, the pH value, ionic conductivity and dissolved oxygen in different electrolytes were probed. As shown in Supplementary Figs. 8 and 9, the AMC dosage did not significantly alter the values, indicating such parameters were unlikely to be the primary drivers behind the observed cation-dependent trend.” in Page 6.

Supplementary Fig. 8 | The pH of electrolytes for 0.5 M H₂SO₄ and 0.5 M H₂SO₄ + 0.3 M A⁺ (A = Li, Na, K, or Cs).

Fig. R3. H₂O₂ yield using Co-CNT at different AMC concentration in a flow-cell reactor.

7. “The addition of AMCs significantly lowered the onset potential and disk current density as compared to the bare electrolyte scenario, which was likely to originate from hindered proton transfer in impeding the HER or 4e⁻ ORR pathways occurring at the catalyst surface^{4,38}” referring to Figure 5b.

In Figure 5b we can see that the ORR onset is, in fact, delayed after the addition of the cations, i.e., the reaction onset happens later in the potential sweep.

Reply 7: Thank you for the detailed comment. The onset of the ORR is indeed shifted to more negative potentials upon the addition of AMCs (Figure 5b). This delayed onset indicates that a higher overpotential is required to initiate the reaction in the presence of AMCs.

We attribute this shift primarily to the **restructuring of the interfacial water network** by AMCs, which hinders proton transport to the catalytic surface. The reduced proton availability raises the kinetic barrier for the initial activation of O₂, thereby delaying the reaction onset. Importantly, this effect preferentially suppresses the 4e⁻ ORR pathway, which relies heavily on efficient proton delivery, while the 2e⁻ pathway is less affected under these conditions. This interpretation is consistent with the statement in our manuscript that “the lowered onset potential and disk current

density likely originate from hindered proton transfer, which impedes both the HER and the dominant $4e^-$ ORR route”.

Thus, the delayed onset not only reflects the overall kinetic modulation induced by AMCs but also provides further evidence for the precise suppression of the $4e^-$ pathway, in good agreement with the enhanced H_2O_2 selectivity observed in the present study.

8. “Our catalyst showcased an outstanding turnover frequency (TOF) value of 0.51 s^{-1} at 0.5 VRHE (Fig. 5d), which compared favorably with reported electrocatalysts^{39,40.}”

Calculating TOF requires knowledge of the precise amount of active sites on the surface. How was this determined and how was it calculated?

Reply 8: Thank you for the careful attention regarding the TOF calculation. As noted, an accurate assessment of the number of active sites is essential for reliable TOF estimation. In our manuscript, the TOF value of 0.51 s^{-1} for the Co-CNT catalyst is derived under the assumption that all cobalt atoms constituting the catalytic cluster architecture function as active sites for H_2O_2 production. The TOF is calculated using the following expression:

$$\text{TOF} = I M_{\text{metal}} / n F m_{\text{catalyst}} w_{\text{metal}}$$

Herein, I (A) represents the Faradaic current for H_2O_2 generation, obtained by multiplying the disk current by the Faradaic efficiency of H_2O_2 . M_{metal} (g/mol) represents the molar mass of cobalt. $n = 2$ corresponds to the number of electrons transferred per H_2O_2 molecule formed. F is the Faraday constant. m_{catalyst} (g) is the total catalyst mass loaded on the electrode, and w_{metal} is the mass fraction of Co in the catalyst, as determined by inductively coupled plasma atomic emission spectroscopy.

The number of active sites is accordingly estimated as $m_{\text{catalyst}} \cdot w_{\text{metal}} / M_{\text{metal}}$. This approach based on the usage of the total metal content as a proxy for the number of active sites, has been **widely employed in the literature for supported metal cluster catalysts**, particularly when precise quantification of exposed surface atoms is experimentally challenging. Similar methodology has been adopted in several recent studies with respect to H_2O_2 electrosynthesis⁵.

It is important to emphasize that, since cobalt is present in the form of clusters rather than fully exposed single atoms, a portion of the Co sites may not be accessible at the electrolyte–catalyst interface. Therefore, the reported TOF of 0.51 s^{-1} should be regarded as a conservative estimate; the actual TOF per exposed active site is likely to be higher. This interpretation is in line with the favorable H_2O_2 production performance observed in our flow-cell tests. To ensure full transparency and reproducibility, we have now expanded the **Methods** section to include a detailed description of the active-site estimation protocol and the underlying assumptions in the revised manuscript.

Revisions to Manuscript:

“This approach, which presumed all Co atoms in the clusters are active, yielded an overestimated count of apparent active site. This, in turn, resulted in a calculated TOF that systematically underestimated the actual catalytic activity.” in Page 25.

9. “To uncover the influence of different electrolyte additives upon catalyst activity, we performed electrochemical double-layer capacitance measurements. As anticipated, the catalyst exhibited the highest electrochemical active surface area value in the Cs⁺-dosed electrolyte, indicative of facilitated mass transport at ORR active sites (Supplementary Fig. 37)43,44”

This is, likely, a misinterpretation. The authors do in fact measure higher capacitance, however, since the cations are participating in the double layer the specific capacitance is likely different in the presence of different cations, while the ECSA is likely the same.

Reply 9: Thank you for the professional comment regarding the electrochemical metrics. We agree with you that there exists **different electrochemical double-layer capacitance with the same ECSA**, owing to the participation of varied cations. Therefore, employing such capacitance value to estimate ECSA trends could introduce quantitative discrepancies if the intrinsic capacitance per unit area varies with cation identity. The capacitive trend analysis, focusing on relative rather than absolute ECSA comparisons, has been utilized in several recent studies addressing cation-mediated electrolyte effects in electrocatalytic systems^{6,7}. To ensure rigorous and unambiguous discussion, **we have revised the relevant section in the revised manuscript to refrain from quantitative ECSA comparison.**

Revisions to Manuscript:

“In addition, electrochemical double-layer capacitance measurements demonstrate that the catalyst exhibits the highest double-layer capacitance in Cs⁺-dosed electrolyte, suggesting an optimized interfacial environment that promotes reaction activity and electron transfer.” in Page 11.

10. *The authors measure the generated H₂O₂ by oxidation at 1.2 V at the ring electrode, but later, when they measure H₂O₂ decomposition (also by oxidation) they perform the measurement at 1V. What is the reason for this? Is the trend the same at 1.2V?*

Reply 10: Thanks for your thoughtful question regarding the selection of applied potentials for H₂O₂ detection and decomposition measurements. The difference in the set-up potentials stems from the **distinct objectives and underlying electrochemistry** of each test.

RRDE measurements: The Pt ring was held at 1.2 V_{RHE} to quantitatively detect H_2O_2 generated at the disk electrode. Although the thermodynamic equilibrium potential for H_2O_2 oxidation to O_2 is approximately 0.695 V_{RHE} , a higher overpotential is required to achieve near 100% detection efficiency due to kinetic limitations. The choice of 1.2 V_{RHE} is an empirical standard that ensures rapid and complete oxidation of H_2O_2 while avoiding interference from the oxygen evolution reaction (OER) at higher potentials.

H_2O_2 reduction reaction ($\text{H}_2\text{O}_2\text{RR}$): $\text{H}_2\text{O}_2\text{RR}$ was evaluated based on LSV profiles on a catalyst-coated glassy carbon electrode in an electrolyte containing 10–50 mM H_2O_2 . Here, the potential range of 0 to 0.8 V_{RHE} is selected to focus on the reduction process ($\text{H}_2\text{O}_2 + 2\text{H}^+ + 2\text{e}^- \rightarrow 2\text{H}_2\text{O}$) while minimizing contributions from H_2O_2 oxidation, which becomes significant above $\sim 1.0 V_{\text{RHE}}$. This approach allows clear observation of the cathodic current associated with H_2O_2 consumption. Regarding the trend at 1.2 V_{RHE} , we note that the oxidation current measured at the ring electrode (at 1.2 V) directly reflects the amount of H_2O_2 produced, and thus the trend in H_2O_2 generation across electrolytes remains consistent. However, if H_2O_2 reduction is to be measured at 1.2 V, the signal would be convoluted by simultaneous oxidation, leading to misinterpretation. Therefore, **the selected potential windows have been optimized for their respective purposes, with the reported trends remaining robust within the applied conditions.**

11. *“The on-site generation of H_2O_2 in our system is highly attractive for targeted applications such as wastewater treatment.”*

How would such high alkaline cation concentrations (0.3M) be applied in practical wastewater treatment. Seems like this would be arguably a worse or equally bad contamination as the contaminants being removed by H_2O_2 generation.

Reply 11: We are grateful for you to raise this critical comment regarding the practical applicability of our electrolyte engineering strategy. The concern about potential secondary contamination from added salts is well-noted and highly relevant for real implementation.

Our systematic investigation demonstrates that AMCs exert a concentration-dependent promotional effect on H_2O_2 electrosynthesis across a wide cation concentration range (0.1–0.5 M), accompanied by a systematic investigation at 0.3 M. Importantly, this concentration range is not only effective but also directly compatible with existing **high-salinity wastewater, particularly reverse osmosis (RO) concentrates**, which typically contain total dissolved solids well above 0.3 M. Following this comment, we have supplemented further discussion in the revised manuscript.

Revisions to Manuscript:

“Our foregoing evaluation confirmed that dosed AMC concentrations ranging from 0.1–0.5 M would enhance H₂O₂ electrosynthesis, with core investigation at 0.3 M that could be encountered in industrial effluents such as reverse osmosis concentrates. Under such conditions, endogenous ions serve as inherent promoters, eliminating the external additives.” in Page 13.

12. Figure 2A the voltammograms are too small to be assessed. Why are the potential windows for the blank and the other CVS different?

Caption for Figure 5f is missing.

Reply 12: Thanks for the valuable suggestions. Following this comment, we have rephrased the voltammogram profile to improve readability and ensure the clear visibility of the key features of diffusion-limited current plateaus (*New Fig. 2a*). The apparent difference in potential windows between the blank and cation-modified systems stems from the marked suppression of proton diffusion upon AMC addition, which shifts the HER onset to more negative potentials. To facilitate direct comparison, we have now aligned all polarization curves within a common potential window in the updated figure. Furthermore, we have included additional comparative LSV curves for each cation system in the revised Supplementary Information file to provide view of the cation-dependent suppression behavior (*New Supplementary Fig. 3*). In addition, we have supplemented the figure caption for Fig. 5f.

Revisions to Manuscript:

Supplementary Fig. 3 | HER polarization curves for Au microelectrodes in Ar-saturated 0.5 M H₂SO₄ + 0.15 M A₂SO₄ at 0.02 V s⁻¹ (A = Li, Na, K, or Cs).

Reviewer #3 (Remarks to the Author):

Authors have presented a comprehensive report on the effects of alkali metal ions on H₂O₂ electrosynthesis and found an interesting size-dependent trend. They have also furnished mechanistic insights into the transformation. While the quantify of experimental work is quite significant, I have some concerns about the impact of the main claims, which renders the current version not par with the Nature Comms level. However, there may be some ways to bridge the gap. Below, I present my major points and invite the co-authors to address them in their revised version, whose suitability I will be happy to advise the Editor on.

Response: Thank you for your approval to our work. All requested revisions based on your instructive comments have been carried out.

Main points:

1. The authors claim that they are the first to investigate the efficacy trend of alkali metal cations (AMC) on the electrosynthesis of H₂O₂, and they suggest that AMC effectiveness follows the trend: Li⁺ < Na⁺ < K⁺ < Cs⁺. This claim seems misleading, as other researchers have investigated the role of AMC on the electrosynthesis of H₂O₂ in acidic media already, and, interestingly, found dissimilar trends.

a. For Instance, Peike Cao et. al. (Reference 14) have also investigated the effect of AMCs and reported that “the effective H₂O₂ currents followed the trend of K⁺ > Na⁺ > Li⁺ ≈ Rb⁺ ≈ Cs⁺ > NH₄⁺, which was related to their hydrated ionic radius,” which seems to contradict the main finding of this manuscript.

b. Also, Zhang et. al. (Reference 13) has also examined the effect of the same (Li⁺, Na⁺, K⁺, Cs⁺) cations on H₂O₂ production and suggested that the “production rates of H₂O₂ are relatively unaffected by the size of the alkali metal cations in the electrolyte.”

These differing views and the underlying factors should be discussed in the manuscript to truly advance the field forward.

Reply 1: Much appreciated for raising this crucial point on the consistency of reported trends for cation-effect. We admit that it is not objective to claim “the first” to report the trend without analyzing the prior studies in this field. In the revision, we have conducted a detailed analysis by fully acknowledging the important work by Cao *et al.* (Ref. [35]) and Zhang *et al.* (Ref. [33]), showing a distinct cation-effect trend.

In further contexts, the core innovation of our work is not merely reporting a cation-effect trend but in providing a **unique mechanistic explanation** rooted in the **cation-specific restructuring of the interfacial hydrogen-bond (H-bond) water network**. Previous studies primarily attributed cation-

effects to electrostatic shielding (Zhang *et al.*) or intermediate stabilization influenced by hydrated ionic radius (Cao *et al.*). Our combined experimental and theoretical evidence reveals that larger cations (Cs^+) more effectively disrupt the interconnected H-bond network at the electrode-electrolyte interface, thereby imposing a greater kinetic barrier to proton transport. The monotonic trend we observe emerges clearly when this interface-specific water restructuring becomes the dominant factor. According to your kind suggestion, we have supplemented further discussion in the revised **Introduction** section.

Revisions to Manuscript:

“Particularly using metal-free carbon-based electrocatalysts, Zhang *et al.* observed that H_2O_2 production rates can be improved by an AMC-induced shielding effect, yet were relatively unaffected by the AMC types³³. A recent study indicated a graphitic carbon electrode exhibited an increase in $2e^-$ ORR selectivity following the trend of $\text{Cs}^+ > \text{K}^+ > \text{Na}^+ > \text{Li}^+$, where different cations would interfere in the $^*\text{OOH}$ binding and influence the local electric field³⁴. It is interesting to note that the AMC roles in acidic $2e^-$ ORR manifest inconsistent trends. In further contexts, the cation effect is effective across other catalyst types apart from carbonaceous materials. For instance, Cao *et al.* found the H_2O_2 production followed the trend of $\text{K}^+ > \text{Na}^+ > \text{Li}^+ \sim \text{Cs}^+$ with the employment of earth-abundant TiC catalyst, which was dictated by the hydrated ionic radius of AMCs³⁵. These disparities imply that the dominant mechanism might be sensitive to specific conditions, such as catalyst identity, cation concentration and operating current density. A coherent picture of how AMCs reshape the interfacial environment in acidic $2e^-$ ORR is still lacking, especially in the realm of prevailing transition metal-involved carbon electrocatalysts (e.g., metal-carbon nanotube, M-CNT).” in Page 3-4.

2. I wish to draw the co-authors' attention to the recent reports on the spontaneous H_2O_2 production at water's interfaces with solids (Refs.1-3), contradicting many groups' experimental and computational claim that the H_2O_2 forms at the air-water interface and has much hitherto unrealized environmental and applied significance (4-12). This specific topic has in fact become very active – a number of reports corroborating the experimental work from the Mishra group (13-17)(also, see <https://arxiv.org/abs/2506.23988> & <https://arxiv.org/abs/2508.15422>. Therefore, these important findings could be highlighted to underscore the importance of this problem (and perhaps an opinion on the right answer!).

Reply 2: Thanks for guiding us to the fascinating and rapidly evolving body of work on **spontaneous H_2O_2 formation at solid-water interfaces** (e.g., Refs. 1-3 & preprints arxiv:2506.23988, arxiv:2508.15422). This is a paradigm-shifting discovery with profound implications for environmental chemistry and catalysis.

While our study focuses on **electrocatalytic** H₂O₂ generation via the 2e⁻ ORR pathway, we fully recognize the significance of these recent reports demonstrating non-electrochemical, **interface-induced spontaneous generation**. This phenomenon underscores the inherent redox activity of water-derived intermediates at confined interfaces, a concept that profoundly enriches the context of our work.

In the revised **Introduction**, we have incorporated a discussion of these findings. We posit that the spontaneous generation and the electrocatalytic generation of H₂O₂ might share a common underlying thread: the critical role of **interfacial water structure and dynamics** in modulating oxygen reduction pathways. In spontaneous processes, unique interfacial water orientations and reactivities may facilitate O₂ activation. In our electrocatalytic system, we demonstrate that alkali metal cations can be used as a “tool” to precisely engineer this interfacial water network, thereby steering the selectivity towards H₂O₂.

Thus, rather than contradicting, these recent discoveries and our work mutually reinforce the overarching importance of **the interfacial aqueous microenvironment** in governing oxygen reduction chemistry. Our study provides a tunable, electrochemical strategy to manipulate this microenvironment for technological gain, directly bridging fundamental interfacial science with applied environmental engineering. We thank you for this insightful connection, which has allowed us to position our work within a broader and more impactful scientific narrative. **We have incorporated the advised references in the revised manuscript (New Ref. [15–26]).**

Revisions to Manuscript:

“Beyond deliberate electrocatalysis, the fundamental chemistry of aqueous interfaces has garnered intense interest for its role in spontaneous chemical transformations¹⁵⁻¹⁸. Notably, the purported spontaneous formation of H₂O₂ has sparked significant debate. While early reports emphasized remarkable accelerations at the air-water interface¹⁹⁻²³, a growing body of evidence, including rigorous work from the Mishra group, suggests that such phenomena might be more pronounced at solid-water interfaces²⁴⁻²⁷. This ongoing controversy underscores that the intrinsic drivers of reactivity in confined aqueous environments remain a fundamental question in interfacial science. Resolving this question is critical not only for understanding natural processes but also for the rational manipulation of electrolyte-catalyst interfaces.” in Page 3.

Added references:

- 15 Eatoo, M. A. *et al.* Why do some metal ions spontaneously form nanoparticles in water microdroplets? Disentangling the contributions of the air–water interface and bulk redox chemistry. *Chem. Sci.* **16**, 1115-1125 (2025).
- 16 Eatoo, M. A. *et al.* Busting the myth of spontaneous formation of H₂O₂ at the air–water

- interface: contributions of the liquid–solid interface and dissolved oxygen exposed. *Chem. Sci.* **15**, 3093-3103 (2024).
- 17 Eatoo, M. A. *et al.* Disentangling the Roles of Dissolved Oxygen, Common Salts, and pH on Spontaneous Hydrogen Peroxide Production in Water: No O₂, No H₂O₂. *J. Am. Chem. Soc.* **147**, 35392-35400, (2025).
- 18 Di Pino, S. *et al.* Deconstructing the origins of interfacial catalysis: Why electric fields are inseparable from solvation. *J. Chem. Phys.* **163**, 184505 (2025).
- 19 Lee, J. K. *et al.* Spontaneous generation of hydrogen peroxide from aqueous microdroplets. *Proc. Natl. Acad. Sci. U. S. A.* **116**, 19294-19298 (2019).
- 20 Mofidfar, M. *et al.* Dependence on relative humidity in the formation of reactive oxygen species in water droplets. *Proc. Natl. Acad. Sci. U. S. A.* **121**, e2315940121 (2024).
- 21 Colussi, A. J. Mechanism of Hydrogen Peroxide Formation on Sprayed Water Microdroplets. *J. Am. Chem. Soc.* **145**, 16315-16317 (2023).
- 22 Angelaki, M. *et al.* Quantification and Mechanistic Investigation of the Spontaneous H₂O₂ Generation at the Interfaces of Salt-Containing Aqueous Droplets. *J. Am. Chem. Soc.* **146**, 8327-8334 (2024).
- 23 Angelaki, M. *et al.* pH Affects the Spontaneous Formation of H₂O₂ at the Air–Water Interfaces. *J. Am. Chem. Soc.* **146**, 25889-25893 (2024).
- 24 Chen, C. J. *et al.* An Alternative Explanation for Ions Put Forth as Evidence for Abundant Hydroxyl Radicals Formed Due to the Intrinsic Electric Field at the Surface of Water Droplets. *Anal. Chem.* **97**, 17687-17695 (2025).
- 25 Gong, K. *et al.* Revisiting the Enhanced Chemical Reactivity in Water Microdroplets: The Case of a Diels–Alder Reaction. *J. Am. Chem. Soc.* **146**, 31585-31596 (2024).
- 26 Asserghine, A. *et al.* Dissolved Oxygen Redox as the Source of Hydrogen Peroxide and Hydroxyl Radical in Sonicated Emulsive Water Microdroplets. *J. Am. Chem. Soc.* **147**, 11851-11858 (2025).

3. In this work, the authors have exploited the potassiumtitanxyoxalate (PTO) reagent for the detection and quantification of H₂O₂. Unfortunately, this reagent is unspecific to H₂O₂, i.e., it can give a false positive signal with reactive oxygen species (ROS) such as hydroxyl radicals, superoxides, and hydroperoxyl radicals, etc. Therefore, the quantification and formation of H₂O₂ should be confirmed by a more reliable and direct technique, such as 1 H-NMR spectroscopy (See Refs.18). It is possible that the reported signals originate from different ROS, and the actual H₂O₂ concentrations are much lower than the reported ones.

Reply 3: Thank you for raising this important methodological point regarding the specificity of the potassium titanyl oxalate (PTO) assay for H₂O₂ quantification. We acknowledge that, in principle,

PTO can react with other reactive oxygen species (ROS), which could potentially lead to an overestimation of H₂O₂ concentration in complex matrices. In the context of our work, several factors assure the reliability of the reported H₂O₂ yields:

System specificity: Our electrochemical system is designed for the selective 2e⁻ ORR to generate H₂O₂. The reaction environment (acidic electrolyte, controlled potential) and the use of a selective Co-CNT catalyst strongly favor the direct production of H₂O₂ over other ROS. The substantial and stable generation of H₂O₂ is further corroborated by its successful on-site application in electro-Fenton reactions for pollutant degradation, which specifically requires H₂O₂ as the precursor for ·OH generation.

¹H-NMR spectroscopy validation: As advised, we have constructed the corresponding standard curve and conducted ¹H-NMR quantification on key samples to provide a validation for the consistency between the potassiumtitanylxalate (PTO) reagent detection and ¹H-NMR spectroscopy method (*New Supplementary Fig. 49*). These results provide strong evidence supporting the accuracy of our reported H₂O₂ production data.

Cross-validation with Ce(IV) titration: To address the concern of assay specificity, we performed parallel quantification using the established cerium(IV) sulfate titration method. As shown in the supplementary figure below (*Fig. R4*), the H₂O₂ concentrations determined by both methods are in good agreement across the measured range. This cross-validation confirms that the signal measured by PTO in our system originates predominantly from H₂O₂.

Widespread use in electrocatalysis tests: The PTO method is a standard, widely adopted technique for H₂O₂ quantification in electrocatalytic studies, including recent high-impact publications, due to its sensitivity, simplicity, and suitability for rapid, high-throughput analysis—a practical necessity for evaluating performance under various reaction conditions. In addition, The PTO method agrees well with ¹H-NMR quantification and cerium(IV) sulfate titration method in our system, demonstrating its practical feasibility.

Supplementary Fig. 49 | Quantification of H₂O₂ by ¹H-NMR spectroscopy. Standard samples of known H₂O₂(aq) concentrations were prepared by diluting a concentrated 30% (v/v) stock solution using HPLC-grade deionized water. All ¹H-NMR tests were carried out at 2°C.

Fig. R4. The linear absorbance-concentration calibration curve of different concentrations of standard H₂O₂ solutions titrated with cerium(IV) sulfate.

4. Lines 76-78: “We observe that the 2e⁻ ORR selectivity in acid follows a monotonically increasing trend from Li⁺ to Cs⁺, which is not revealed by prior studies (Figure 1).” Figure 1 is an illustration; it does not represent any experimental observation.

Reply 4: Much appreciated for this detailed suggestion. Figure 1 is a conceptual scheme designed to summarize the proposed mechanism of cation-mediated interfacial water restructuring, which is not suitable to represent the experimental trend as you pointed out. Following this comment, we have **repositioned Figure 1 in the revised manuscript** to ensure a clear distinction between the presentation of experimental data (in subsequent figures) and explanatory scheme.

Revisions to Manuscript:

“The varying influences of different AMCs stem from their distinct capabilities to perturb the H-bond network, associated with their hydration properties and coordination strengths. Such a perspective is essentially different from the general recognition of AMC roles in this field (Fig. 1).” in Page 4.

5. *Supplementary Fig. 3: Concentration-dependent variation of suppression effect—How does the concentration of electrolyte affect the solubility of oxygen in the electrolyte? It is essential to quantify the dissolved oxygen concentrations in the electrolyte as a function of the electrolyte*

concentration. Some other important parameters that need to be kept comparable are ionic conductivity and dissolved oxygen levels.

Reply 5: Thank you for raising this critical point. We fully agree that these are essential variables to control and characterize when comparing electrochemical performance across different electrolytes, as they exert direct impact upon mass transport and reaction kinetics.

Following this comment, we have supplemented systematic measurements to quantify these key parameters in all electrolyte systems used in this study (0.5 M H₂SO₄ + 0.15 M Li₂SO₄, Na₂SO₄, K₂SO₄ or Cs₂SO₄). The relating ionic conductivity and dissolved oxygen concentrations results have been summarized in the *New Supplementary Fig. 9*.

Ionic conductivity: Conductivity was measured using a calibrated conductivity meter at 25 °C. The addition of 0.3 M AMC caused only a marginal variation in overall conductivity (167–173 mS cm⁻¹) compared to the baseline 0.5 M H₂SO₄ (~168 mS cm⁻¹). This minor variation is unlikely to be the primary driver of the cation-specific trends observed in ORR selectivity and activity.

Dissolved oxygen concentration: Dissolved oxygen levels were quantified using a Clark-type oxygen electrode in O₂-saturated electrolytes at 25°C. The data confirm that for a given total molarity of added sulfate (e.g., 0.15 M A₂SO₄), the dissolved oxygen concentrations were nearly identical (± 3%) across the different AMCs (Li⁺, Na⁺, K⁺, or Cs⁺). This indicates that the relative comparison of cation effects presented in our work (e.g., the trend Li⁺ < Na⁺ < K⁺ < Cs⁺) was performed under effectively equivalent O₂ availability.

These supplemental data confirm that the monotonic cation-dependent trends in 2e⁻ ORR selectivity and H₂O₂ production rate are attributed to the specific cationic modulation of the interfacial water structure and proton transport.

Revisions to Manuscript:

“To exclude potential interference from other variables, the pH value, ionic conductivity and dissolved oxygen in different electrolytes were probed. As shown in Supplementary Figs. 8 and 9, the AMC dosage did not significantly alter the values, indicating such parameters were unlikely to be the primary drivers behind the observed cation-dependent trend. These results motivated us to investigate the restructuring of interfacial H-bonding water networks in the presence of AMCs.” in Page 6.

Supplementary Fig. 9 | (a) Electrolyte ionic conductivity (EC) and (b) electrolyte dissolved oxygen (DO) for 0.5 M H₂SO₄ and 0.5 M H₂SO₄ + 0.3 M A⁺ (A = Li, Na, K, or Cs).

6. Use the right symbol for free radicals: replace *OOH, *OH, *O by •OOH, •OH, •O (Asterisk by Dot).

Reply 6: We sincerely thank you for the meticulous attention to notation. Indeed, the asterisk * explicitly represents an active site on the catalyst, which is distinct from the notation for free radicals in solution (e.g., •OH, •OOH) employing a dot (•) to indicate an unpaired electron. To avoid any ambiguity, we have carefully reviewed the entire manuscript to ensure this notation is used correctly and consistently.

“*OOH” in Page 9-14, “*OH” in Page 9, “*O” in Page 9,14, “•OH” in Page 13.

Employed Reference:

- 1 Saerom Yu. *et al.* What Is the Rate-Limiting Step of Oxygen Reduction Reaction on Fe–N–C Catalysts? *J. Am. Chem. Soc.* **145**, 25352-25356 (2023).
- 2 X. Zhao. *et al.* Origin of Selective Production of Hydrogen Peroxide by Electrochemical Oxygen Reduction. *J. Am. Chem. Soc.* **143**, 9423-9428(2021).
- 3 Y. Liu. *et al.* Monomolecule Coupled to Oxygen-Doped Carbon for Efficient Electrocatalytic Hydrogen Peroxide Production. *Adv. Mater.* **37**, 2502197(2025).
- 4 K. Yu *et al.* Engineering Asymmetric Electronic Structure of Co–N–C Single-Atomic Sites Toward Excellent Electrochemical H₂O₂ Production and Biomass Upgrading *Angew. Chem. Int. Ed.* **64** e202502383(2025).
- 5 Q. An. *et al.* In Situ Identification of Zinc Sites as Potential-Dependent Selectivity Switch over Dual-Atom Catalysts for H₂O₂ Electrosynthesis. *J. Am. Chem. Soc.* **147**, 11465-11476(2025).
- 6 Gu, J. *et al.* Modulating electric field distribution by alkali cations for CO₂ electroreduction

in strongly acidic medium. *Nat. Catal.* **5**, 268–276(2022).

- 7 B. Huang. *et al.* Cation- and pH-Dependent Hydrogen Evolution and Oxidation Reaction Kinetics. *JACS Au* **1**, 1674-1687(2021).

Response to Reviewers' Comments (NCOMMS-25-75578A)

Reviewer #1 (Remarks to the Author):

The revision has addressed most of my previous concerns and the manuscript is now close to acceptance. I believe the work can be accepted after the authors clarify the following minor points to improve transparency and reproducibility.

Response: Thank you for your approval to our work. All requested revisions based on your detailed comments have been carried out.

Specific comment:

1. In the rebuttal, the authors state that adsorption energies of alkali metal cations (AMCs) on the catalyst surface were calculated to exclude direct surface binding. It would be helpful to explicitly clarify whether the positive charge state of the cations (e.g., $\text{Li}^+/\text{Na}^+/\text{K}^+$) was considered in these calculations, and how charge compensation was handled in the computational model (e.g., background charge, counter ions, solvation treatment, etc.), since this may influence adsorption energetics and interfacial electrostatics.

Reply 1: Thank you for your inspiring suggestions regarding the theoretical calculation of charged species. In the adsorption energy calculations for AMCs (Li^+ , Na^+ , K^+ , Cs^+) on the catalyst surface, we explicitly accounted for their charge state. This was done by manually adjusting the number of valence electrons in the computational model to reflect the cationic charge, consistent with standard practice for simulating charged adsorbates in periodic calculations.

To maintain the interfacial electrostatic environment in actual experiments, we adopted the following strategy in the simulation:

Charge compensation: A uniform background charge (jellium model) was applied to neutralize the system, which is a common and efficient approach for periodic DFT calculations involving isolated charged species.

Solvation treatment: The implicit solvation model (VASPsol++) was employed throughout, which self-consistently accounts for the dielectric response of the electrolyte and helps stabilize the charged interface.

Model validation: The adsorption energies obtained under this setup consistently showed positive values for all AMCs, confirming that the cations remain preferentially solvated rather than adsorbing directly onto the catalyst surface, which is consistent with our experimental interpretation.

These methodological details, including the handling of cation charge states and the charge compensation scheme, have now been incorporated into the *Methods* section of the revised manuscript (please see “DFT Calculations” subsection) to ensure full transparency and reproducibility.

Revisions to Manuscript (in Page 27):

“We applied a constant potential (+ 0.15 V vs SHE, consistent with experiments) to the ORR simulations by CP-VASP.^{90,91} The targeted Fermi level referenced to the electrolyte was set to -4.75 V with a convergence criterion of 0.01 V for the electrolyte-referenced Fermi level. The solvation model during in calculations was considered by using the VASPsol++ framework implemented in VASP.⁹²

Each AMC was put on the surface of Co-CNT and the adsorption energies were calculated by DFT:

$$E_{ad} = E_{total} - E_{Co-CNT} - E_{AMC}$$

where E_{total} is the total energy of an AMC adsorbed on the catalyst, while E_{Co-CNT} and E_{AMC} represent the energies of Co-CNT and AMC, separately. Positive-charged calculations were performed by manually setting the number of valence electrons to consider the positive charge state of AMCs.”

2. *The authors performed constant-potential DFT calculations at 0.15 V vs. SHE to account for electrochemical conditions. While this value is described as being consistent with experimental conditions, it would strengthen the manuscript to provide a clearer electrochemical/theoretical rationale for selecting this potential, or to briefly discuss how representative it is within the relevant ORR operating window and whether the conclusions are expected to remain robust over a reasonable potential range.*

Reply 2: We thank the reviewer for this thoughtful question regarding the choice of the applied potential in our constant-potential DFT calculations. The potential of +0.15 V vs. SHE was selected because it corresponds directly to the working electrode potential measured during our flow-cell experiments at a current density of 50 mA cm⁻², a commonly employed current density in laboratory-scale studies of H₂O₂ electrosynthesis (Ref: *J. Am. Chem. Soc.* **2024**, 146, 9434; *Adv. Funct. Mater.* **2025**, 35, 2417090).

A brief discussion of the potential selection and the associated validation have now been added to the revised manuscript (*Methods* section).

Revisions to Manuscript (in Page 27):

“Each AMC was put on the surface of Co-CNT and the adsorption energies were calculated by DFT:

$$E_{ad} = E_{total} - E_{Co-CNT} - E_{AMC}$$

where E_{total} is the total energy of an AMC adsorbed on the catalyst, while E_{Co-CNT} and E_{AMC} represent the energies of Co-CNT and AMC, separately. Positive-charged calculations were performed by manually setting the number of valence electrons to consider the positive charge state of AMCs.”

3. In the *Methods* section, particularly in the “DFT calculations” subsection near the end of the manuscript, there are currently no references cited for the computational methods and models employed. Please add appropriate methodological references (e.g., DFT setup, electrochemical/solvation model, constant-potential method) to improve completeness and reproducibility.

Reply 3: We apologize for the oversight in not including the methodological references in the original submission. We have now carefully added all relevant references in the “DFT Calculations” subsection of the *Methods* section. These include citations for:

- (1) The DFT software package and computational setup (VASP, PAW pseudopotentials, plane-wave cutoff, exchange-correlation functional, and dispersion correction).
- (2) The implicit solvation model (VASPsol++) used to describe the electrolyte environment.
- (3) The constant-potential (CP-VASP) methodology employed to incorporate the electrochemical potential in our ORR simulations.
- (4) Key methodological papers supporting the thermodynamic analysis (e.g., free-energy corrections, treatment of adsorbates, and determination of reaction pathways).

By providing these references, we aim to offer a fully transparent and reproducible account of our computational approach. The related references have been updated as **New Ref. [70-93]**.

Revisions to Manuscript (Added References):

70. Berendsen, H. J. C. et al. A message-passing parallel molecular-dynamics implementation. *Comput. Phys. Commun.* **91**, 43–56 (1995).
71. Lindahl, E. et al. GROMACS 3.0: A package for molecular simulation and trajectory analysis. *J. Mol. Model.* **7**, 306–317 (2001).
72. Hess, B. et al. GROMACS 4: Algorithms for highly efficient, load-balanced, and scalable molecular simulation. *J. Chem. Theory Comput.* **4**, 435–447 (2008).
73. Berendsen, H. J. C. et al. Molecular dynamics with coupling to an external bath. *J. Chem. Phys.* **81**, 3684–3690 (1984).
74. Darden, T. et al. Particle mesh Ewald: An N log(N) method for Ewald sums in large systems. *J. Chem. Phys.* **98**, 10089–10092 (1993).
75. Markland, T. E. et al. Nuclear quantum effects enter the mainstream. *Nat. Rev. Chem.* **2**, 0109 (2018).

76. Rebstock, J. A. et al. Comparing interfacial cation hydration at catalytic active sites and spectator sites on gold electrodes. *Chem. Sci.* **13**, 7634–7643 (2022).
77. Ringe, S. et al. Understanding cation effects in electrochemical CO₂ reduction. *Energy Environ. Sci.* **12**, 3001–3014 (2019).
78. Rossi, M. et al. Nuclear quantum effects in H⁺ and OH⁻ diffusion along confined water wires. *J. Phys. Chem. Lett.* **7**, 3001–3007 (2016).
79. Varma, S. et al. Coordination numbers of alkali metal ions in aqueous solutions. *Biophys. Chem.* **124**, 192–199 (2006).
80. Kühne, T. D. et al. CP2K: An electronic structure and molecular dynamics software package—Quickstep. *J. Chem. Phys.* **152**, 194103 (2020).
81. Ozkanlar, A. et al. ChemNetworks: A complex network analysis tool for chemical systems. *J. Comput. Chem.* **35**, 495–505 (2014).
82. Kresse, G. et al. Ab initio molecular-dynamics simulation of the liquid–metal amorphous–semiconductor transition in germanium. *Phys. Rev. B* **49**, 14251–14269 (1994).
83. Kresse, G. Ab initio molecular dynamics for liquid metals. *J. Non-Cryst. Solids* **193**, 222–229 (1995).
84. Kresse, G. et al. Efficiency of ab initio total energy calculations for metals and semiconductors using a plane-wave basis set. *Comput. Mater. Sci.* **6**, 15–50 (1996).
85. Kresse, G. et al. Efficient iterative schemes for ab initio total-energy calculations using a plane-wave basis set. *Phys. Rev. B* **54**, 11169–11186 (1996).
86. Kresse, G. et al. From ultrasoft pseudopotentials to the projector augmented-wave method. *Phys. Rev. B* **59**, 1758–1775 (1999).
87. Perdew, J. P. et al. Atoms, molecules, solids, and surfaces: Applications of the generalized gradient approximation for exchange and correlation. *Phys. Rev. B* **48**, 4978–4978 (1993).
88. Hammer, B. et al. Improved adsorption energetics within density-functional theory using revised Perdew–Burke–Ernzerhof functionals. *Phys. Rev. B* **59**, 7413–7421 (1999).
89. Grimme, S. et al. Effect of the damping function in dispersion-corrected density functional theory. *J. Comput. Chem.* **32**, 1456–1465 (2011).
90. Zhao, X. H. et al. Origin of selective production of hydrogen peroxide by electrochemical oxygen reduction. *J. Am. Chem. Soc.* **143**, 9423–9428 (2021).
91. Yu, S. R. et al. What is the rate-limiting step of oxygen reduction reaction on Fe–N–C catalysts? *J. Am. Chem. Soc.* **145**, 25352–25356 (2023).
92. Islam, S. M. R. et al. An implicit electrolyte model for plane wave density functional theory exhibiting nonlinear response and a nonlocal cavity definition. *J. Chem. Phys.* **159**, 204103 (2023).
93. Bochevarov, A. D. et al. Jaguar: A high-performance quantum chemistry software program with strengths in life and materials sciences. *Int. J. Quantum Chem.* **113**, 2110–2142 (2013).

Reviewer #2 (Remarks to the Author):

Dear Authors, Dear Editors,

I feel like all my remarks have been answered appropriately.

Best regards.

Response: Thank you for your approval of our work.

Reviewer #3 (Remarks to the Author):

I am satisfied by the additional work the co-authors have done to elevate their contribute to Nature Comms. level. I recommend it's acceptance.

Response: Thank you for your approval of our work.

Title: Differentiating interfacial water structures via alkali metal cation promotor for H₂O₂ electrosynthesis in acid

Authors: Yifei Wang^{1*}, Peiyang Duan¹, Yingqi Liao², Hao Wang¹, Beibei Li¹, Hangyuan Zhang^{3, 5}, Hao Yang^{2*}, Tao Cheng^{2*}, and Jingyu Sun^{3*}

Summary: Authors have presented a comprehensive report on the effects of alkali metal ions on H₂O₂ electrosynthesis and found an interesting size-dependent trend. They have also furnished mechanistic insights into the transformation. While the quantify of experimental work is quite significant, I have some concerns about the impact of the main claims, which renders the current version not par with the *Nature Comms* level. However, there may be some ways to bridge the gap. Below, I present my major points and invite the co-authors to address them in their revised version, whose suitability I will be happy to advise the Editor on.

Main points:

1. The authors claim that they are the first to investigate the efficacy trend of alkali metal cations (AMC) on the electrosynthesis of H₂O₂, and they suggest that AMC effectiveness follows the trend: $\text{Li}^+ < \text{Na}^+ < \text{K}^+ < \text{Cs}^+$. This claim seems misleading, as other researchers have investigated the role of AMC on the electrosynthesis of H₂O₂ in acidic media already, and, interestingly, found dissimilar trends.
 - a. For Instance, Peike Cao et. al. (Reference 14) have also investigated the effect of AMCs and reported that “*the effective H₂O₂ currents followed the trend of $\text{K}^+ > \text{Na}^+ > \text{Li}^+ \approx \text{Rb}^+ \approx \text{Cs}^+ > \text{NH}_4^+$, which was related to their hydrated ionic radius,*” which seems to contradict the main finding of this manuscript.
 - b. Also, Zhang et. al. (Reference 13) has also examined the effect of the same (Li^+ , Na^+ , K^+ , Cs^+) cations on H₂O₂ production and suggested that the “*production rates of H₂O₂ are relatively unaffected by the size of the alkali metal cations in the electrolyte.*”

These differing views and the underlying factors should be discussed in the manuscript to truly advance the field forward.

2. I wish to draw the co-authors' attention to the recent reports on the spontaneous H₂O₂ production at water's interfaces with solids (Refs.¹⁻³), contradicting many groups' experimental and computational claim that the H₂O₂ forms at the air-water interface and has much hitherto unrealized environmental and applied significance⁴⁻¹². This specific topic has in fact become very active – a number of reports corroborating the experimental work from the Mishra group¹³⁻¹⁷ (also, see <https://arxiv.org/abs/2506.23988> & <https://arxiv.org/abs/2508.15422>). Therefore, these important findings could be highlighted to underscore the importance of this problem (and perhaps an opinion on the right answer!).
3. In this work, the authors have exploited the potassiumtitanoyloxalate (PTO) reagent for the detection and quantification of H₂O₂. Unfortunately, this reagent unspecific to H₂O₂, i.e., it can give a false positive signal with reactive oxygen species (ROS) such

Field Code Changed

as hydroxyl radicals, superoxides, and hydroperoxyl radicals, etc. Therefore, the quantification and formation of H₂O₂ should be confirmed by a more reliable and direct technique, such as ¹H-NMR spectroscopy (See Refs.¹⁸). It is possible that the reported signals originate from different ROS, and the actual H₂O₂ concentrations are much lower than the reported ones.

4. Lines 76-78: “We observe that the 2e⁻ ORR selectivity in acid follows a monotonically increasing trend from Li⁺ to Cs⁺, which is not revealed by prior studies (Figure 1).” **Figure 1** is an illustration; it does not represent any experimental observation.
5. Supplementary Fig. 3: Concentration-dependent variation of suppression effect—How does the concentration of electrolyte affect the solubility of oxygen in the electrolyte? It is essential to quantify the dissolved oxygen concentrations in the electrolyte as a function of the electrolyte concentration. Some other important parameters that need to be kept comparable are ionic conductivity and dissolved oxygen levels.
6. Use the right symbol for free radicals: replace *OOH, *OH, *O by •OOH, •OH, •O (Asterisk by Dot).

References:

- 1 Eatoo, M. A., Wehbe, N., Kharbatia, N., Guo, X. & Mishra, H. Why do some metal ions spontaneously form nanoparticles in water microdroplets? Disentangling the contributions of the air-water interface and bulk redox chemistry. *Chem Sci* **16**, 1115-1125, doi:10.1039/d4sc03217a (2025).
- 2 Eatoo, M. A. & Mishra, H. Busting the myth of spontaneous formation of H₂O₂ at the air-water interface: contributions of the liquid-solid interface and dissolved oxygen exposed. *Chem Sci* **15**, 3093-3103, doi:10.1039/d3sc06534k (2024).
- 3 Eatoo, M. A. & Mishra, H. Disentangling the Roles of Dissolved Oxygen, Common Salts, and pH on Spontaneous Hydrogen Peroxide Production in Water: No O₂, No H₂O₂. *J Am Chem Soc* **147**, 35392-35400, doi:10.1021/jacs.5c09028 (2025).
- 4 Lee, J. K. *et al.* Spontaneous generation of hydrogen peroxide from aqueous microdroplets. *Proceedings of the National Academy of Sciences of the United States of America* **116**, 19294-19298, doi:10.1073/pnas.1911883116 (2019).
- 5 Lee, J. K. *et al.* Condensing water vapor to droplets generates hydrogen peroxide. *Proceedings of the National Academy of Sciences of the United States of America* **117**, 30934-30941, doi:10.1073/pnas.2020158117 (2020).
- 6 Mehrgardi, M. A., Mofidfar, M. & Zare, R. N. Sprayed Water Microdroplets Are Able to Generate Hydrogen Peroxide Spontaneously. *J Am Chem Soc* **144**, 7606-7609, doi:10.1021/jacs.2c02890 (2022).
- 7 Mofidfar, M., Mehrgardi, M. A., Xia, Y. & Zare, R. N. Dependence on relative humidity in the formation of reactive oxygen species in water droplets. *Proc Natl Acad Sci U S A* **121**, e2315940121, doi:10.1073/pnas.2315940121 (2024).

- 8 Xing, D. *et al.* Challenges in Detecting Hydroxyl Radicals Generated in Water Droplets with Mass Spectrometry. *Anal Chem*, doi:10.1021/acs.analchem.5c00386 (2025).
- 9 Heindel, J. P., Hao, H., LaCour, R. A. & Head-Gordon, T. Spontaneous formation of hydrogen peroxide in water microdroplets. *The Journal of Physical Chemistry Letters* **13**, 10035-10041 (2022).
- 10 Colussi, A. J. Mechanism of Hydrogen Peroxide Formation on Sprayed Water Microdroplets. *J Am Chem Soc* **145**, 16315-16317, doi:10.1021/jacs.3c04643 (2023).
- 11 Angelaki, M., Carreira Mendes Da Silva, Y., Perrier, S. & George, C. Quantification and Mechanistic Investigation of the Spontaneous H₂O₂ Generation at the Interfaces of Salt-Containing Aqueous Droplets. *J Am Chem Soc* **146**, 8327-8334, doi:10.1021/jacs.3c14040 (2024).
- 12 Angelaki, M., d'Erceville, J., Donaldson, D. J. & George, C. pH Affects the Spontaneous Formation of H₂O₂ at the Air-Water Interfaces. *J Am Chem Soc* **146**, 25889-25893, doi:10.1021/jacs.4c07356 (2024).
- 13 Chen, C. J. & Williams, E. R. A Source of the Mysterious m/z 36 Ions Identified: Implications for the Stability of Water and Unusual Chemistry in Microdroplets. *ACS Central Science*, doi:10.1021/acscentsci.5c00306 (2025).
- 14 Chen, C. J. & Williams, E. R. An Alternative Explanation for Ions Put Forth as Evidence for Abundant Hydroxyl Radicals Formed Due to the Intrinsic Electric Field at the Surface of Water Droplets. *Analytical Chemistry*, doi:10.1021/acs.analchem.5c02973 (2025).
- 15 Koppenol, W. H. & Sies, H. Was hydrogen peroxide present before the arrival of oxygenic photosynthesis? The important role of iron(II) in the Archean ocean. *Redox Biology* **69**, doi:ARTN 103012
10.1016/j.redox.2023.103012 (2024).
- 16 Gong, K. *et al.* Revisiting the Enhanced Chemical Reactivity in Water Microdroplets: The Case of a Diels–Alder Reaction. *J Am Chem Soc* **146**, 31585-31596, doi:10.1021/jacs.4c09400 (2024).
- 17 Asserghine, A. *et al.* Dissolved Oxygen Redox as the Source of Hydrogen Peroxide and Hydroxyl Radical in Sonicated Emulsive Water Microdroplets. *J Am Chem Soc* **147**, 11851-11858, doi:10.1021/jacs.4c16759 (2025).
- 18 Kakeshpour, T. & Bax, A. NMR characterization of H₂O₂ hydrogen exchange. *Journal of Magnetic Resonance* **333**, 107092, doi:<https://doi.org/10.1016/j.jmr.2021.107092> (2021).